# Adventitial fibroblasts direct smooth muscle cell-state transition in pulmonary vascular disease

Slaven Crnkovic[1,2,3], Helene Thekkekara Puthenparampil[1], Shirin Mulch[3], Valentina Biasin[1,2], Nemanja Radic[2], Jochen Wilhelm[3], Marek Bartkuhn[3], Ehsan Bonyadi Rad[1], Alicja Wawrzen[2], Ingrid Matzer[2], Ankita Mitra[4], Ryan D Leib[5], Bence Miklos Nagy[1], Anita Sahu-Osen[1], Francesco Valzano[1], Natalie Bordag[1,2], Matthias Evermann[6], Konrad Hoetzenecker[6], Andrea Olschewski[1,2], Senka Ljubojevic-Holzer[2], Malgorzata Wygrecka[3], Kurt Stenmark[7], Leigh M Marsh[1,2], Vinicio de Jesus Perez[4], Grazyna Kwapiszewska[1,2,3]*

[1]Ludwig Boltzmann Institute for Lung Vascular Research, Graz, Austria; [2]Medical University of Graz, Lung Research Cluster, Graz, Austria; [3]Institute for Lung Health, Cardiopulmonary Institute, Member of the German Center for Lung Research, Justus-Liebig University Giessen, Giessen, Germany; [4]Department of Medicine, Stanford University School of Medicine, Stanford, United States; [5]Mass Spectrometry Laboratory, Stanford University School of Medicine, Stanford, United States; [6]Medical University of Vienna, Vienna, Austria; [7]Developmental Lung Biology and Cardiovascular Pulmonary Research Laboratories, University of Colorado, Aurora, United States

*For correspondence: grazyna.kwapiszewska-marsh@ medunigraz.at

Competing interest: The authors declare that no competing interests exist.

## eLife Assessment

This **fundamental** research conducted a molecular comparison between smooth muscle cells and adjacent fibroblast cells within lung blood vessels affected by pulmonary arterial hypertension. The study identified distinct disease-related states in each cell type and provided deeper insights into their interactions and communication. While certain conclusions should be interpreted with caution due to inherent methodological limitations, the study's findings remain **convincing** and robust. This is supported by the use of advanced and complementary techniques, as well as the rare isolation of diseased lung blood vessel cells from the same donor, enabling direct comparison.

## Abstract

**Background:** Pulmonary vascular remodeling is a progressive pathological process characterized by functional alterations within pulmonary artery smooth muscle cells (PASMCs) and adventitial fibroblasts (PAAFs). Mechanisms driving the transition to a diseased phenotype remain elusive.

**Methods:** We combined transcriptomic and proteomic profiling with phenotypic characterization of source-matched cells from healthy controls and individuals with idiopathic pulmonary arterial hypertension (IPAH). Bidirectional cellular crosstalk was examined using direct and indirect co-culture models, and phenotypic responses were assessed via transcriptome analysis.

**Results:** PASMC and PAAF undergo distinct phenotypic shifts during pulmonary vascular remodeling, with limited shared features, such as reduced mitochondrial content and hyperpolarization. IPAH-PASMC exhibit increased glycosaminoglycan production and downregulation of contractile machinery, while IPAH-PAAF display a hyperproliferative phenotype. We identified alterations in

extracellular matrix components, including laminin and collagen, alongside pentraxin-3 and hepatocyte growth factor, as potential regulators of PASMC phenotypic transitions mediated by PAAF.
**Conclusions:** While PASMCs and PAAFs retain their core cellular identities, they acquire distinct disease-associated states. These findings provide new insights into the dynamic interplay of pulmonary vascular mesenchymal cells in disease pathogenesis.

**Funding:** This work was supported by Cardio-Pulmonary Institute EXC 2026 390649896 (GK) and Austrian Science Fund (FWF) grant I 4651-B (SC).

## Introduction

Classical view of vascular smooth muscle cells as resident cells mediating vasoactive responses and adventitial fibroblasts as structural support cells producing fibrillar collagen has been appended by single-cell transcriptomics studies in humans. Single-cell resolution has helped to define the tissue-specific heterogeneity within and delineate the distinctions of vascular resident cell populations to other smooth muscle and fibroblast populations (*Crnkovic et al., 2022*; *Muhl et al., 2022*; *Muhl et al., 2020*; *Travaglini et al., 2020*). Additional layer of cellular complexity and heterogeneity is acquired through the disease process. Fate mapping studies in animal models of systemic and pulmonary vascular disease have shown that the majority of vascular smooth muscle cells and adventitial fibroblasts retain their lineage identity during vascular remodeling (*Biasin et al., 2020*; *Bordenave et al., 2020*; *Sheikh et al., 2015*; *Steffes et al., 2020*; *Wirka et al., 2019*; *Worssam et al., 2023*). However, a subset of these cells exhibits the acquisition or loss of specific markers, such as smooth muscle actin (*Chu et al., 2024*; *Shankman et al., 2015*; *Short et al., 2004*). Nevertheless, diseased cells consistently demonstrate altered expression profiles compared to their healthy counterparts (*Crnkovic et al., 2022*; *Wirka et al., 2019*; *Worssam et al., 2023*; *Zhang et al., 2024*).

Pulmonary vascular disease is an umbrella term encompassing functional and morphological alterations in lung vasculature, either as a standalone condition or, more often, accompanying chronic lung and heart disease (*Humbert et al., 2019*). We have previously shown that pulmonary vascular disease and remodeling of pulmonary artery compartment is associated with a skewed cellular communication centered on smooth muscle cells and adventitial fibroblasts (*Crnkovic et al., 2022*). Moreover, molecular alterations identified in situ and freshly isolated cells are at least partially preserved and propagated in vitro. For example, pulmonary artery smooth muscle cells (PASMCs) and pulmonary artery adventitial fibroblasts (PAAFs) isolated from remodeled pulmonary arteries display metabolic and proinflammatory reprogramming that was linked with the progression of pulmonary vascular remodeling (*Hong et al., 2017*; *Li et al., 2023*; *Sutendra et al., 2010*). On transcriptional and proteomic level, PASMC displays characteristics of phenotypic shift, namely downregulated expression of contractile machinery and upregulation of extracellular matrix (ECM) components (*Crnkovic et al., 2022*; *Régent et al., 2016*). PAAF was shown to direct the behavior of immune cells and serving as a homing place for the immune cells during vascular remodeling (*Chelladurai et al., 2022*; *El Kasmi et al., 2014*; *Plecitá Hlavatá et al., 2023*). Cumulatively, results imply a cell-type-specific communication network that is perturbed in the vascular remodeling. It is however unclear to which extent are the changes in expression profiles ultimately mirrored phenotypically.

In the current study, we investigated early-passage PASMC and PAAF sourced from the same pulmonary artery of healthy, downsized lung donors or patients with idiopathic form of pulmonary arterial hypertension (IPAH). We have combined integrative omics and network-based analysis with targeted phenotypic screens to identify common and cell-type-specific differences. Our aim was to uncover significant phenotypic differences upon pulmonary vascular remodeling and identify a set of key molecular correlates that could serve as surrogate marker defining healthy and diseased cell states in a cell-type-specific manner. Our results provide insight into bidirectional cell communication between PASMC and PAAF and the critical role of heterotypic cell-to-cell interaction in cell-state maintenance and transition in pulmonary artery compartment.

**eLife digest** Idiopathic pulmonary hypertension (IPAH for short) is a chronic disease that can lead to severe breathing difficulties and a higher risk of heart damage or other life-threatening complications. While drugs exist to help manage symptoms, none are currently curative.

IPAH mainly affects the pulmonary arteries, which are tasked with carrying the blood from the heart to the lungs to collect oxygen. The disease causes the walls of these vessels to thicken abnormally, obstructing blood flow and requiring the heart to work harder.

The outer layers of pulmonary arteries are formed of both fibroblast and smooth muscle cells: the fibroblasts produce proteins that support the structure of the vessel, while the smooth muscle helps control blood flow by contracting and relaxing. Both types of cells are involved in vessel wall thickening in IPAH, but what causes them to switch from a healthy to a diseased state remains unclear.

To examine this question, Crnkovic et al. compared the gene activity, protein production and behaviour of smooth muscle cells and fibroblasts obtained from the pulmonary arteries of healthy donors and IPAH patients. The experiments revealed that both types of cells showed similar impairments to their mitochondria, the cellular structures that help provide energy. However, differences also emerged. Diseased smooth muscle tissue produced the proteins that these cells use to contract and relax, ultimately impairing their ability to regulate blood flow to the lungs. Meanwhile, IPAH fibroblasts multiplied abnormally fast, potentially contributing to vessel thickening.

Growing smooth muscle and fibroblast cells together in the laboratory allowed Crnkovic et al. to examine how interactions between these cells could help drive the disease. This showed that the fibroblasts released signals which triggered IPAH-characteristic changes in normally healthy smooth muscle cells. As such, the ability of fibroblasts to 'talk' to smooth muscle cells could be important in the progression of the condition.

Crnkovic et al. hope that these findings will help develop better treatments to stop or reverse the progression of IPAH. However, further research is needed to confirm which biological processes should be targeted, and to ensure that reversing the abnormal changes in lung blood vessels is both safe and effective.

## Methods

### Human tissue samples

Human lung samples from IPAH patients and downsized non-transplanted donor lungs, serving as healthy control, were obtained from Division of Thoracic Surgery, Medical University of Vienna, Austria. The protocol and tissue usage was approved by the local ethics committee (976/2010; 1417/2022) and patient consent was obtained before lung transplantation. Small size, resistance pulmonary arteries (diameter <0.5 cm) were dissected from the lungs and used for smooth muscle cell and fibroblast cell isolation or flash-frozen in liquid nitrogen. Briefly, adventitial layer was pulled off the PA and used for outgrowth of PAAF. Remaining PA was cut open, endothelial layer scraped off and remaining media mechanically cut into small size pieces and used for smooth muscle cell outgrowth. Cells were grown in full media (VascuLife SMC or FibroLife S2, LifeLine Cell Technology) and used up to passage four. Respective basal media without added supplements and serum was used for starvation. Expression profiling (transcriptomic and proteomic) and functional cellular assays were performed on cells isolated from the same pulmonary artery. Histological staining of formalin-fixed paraffin-embedded human lung samples ($n$ = 5–10) included same control/patient cases that were used for in vitro studies. Frozen PA used for glysoaminoglycan measurement ($n$ = 8 for each condition) consisted of an independent control/patient cohort. Clinical data are given in *Supplementary file 1*.

### Genome-wide expression profiling

Briefly, whole-genome expression profiling was performed on total RNA isolated using RNeasy Mini kit (Peqlab) from PAAF and PASMC in first passage (four healthy donors and four IPAHs). Purified total RNA was amplified and Cy3-labeled using the LIRAK kit (Agilent) following the kit instructions. Per reaction, 200 ng of total RNA was used. The SureTag DNA labeling kit (Agilent, Waldbronn, Germany) was used to Cy5 and Cy3 labels the samples and subsequently hybridized to 60-mer oligonucleotide

spotted microarray slides (Agilent Human G3 8x60k, Design ID 072363). The following hybridization, washing, and drying steps were performed following the Agilent hybridization protocol. Thereafter, the slides were scanned at 2 µm/pixel resolution using the InnoScan 900 (Innopsys, Carbonne, France). Image analysis was performed with Mapix 6.5.0 software, and calculated values for all spots were saved as GenePix results files. Stored data were evaluated using the R software and the limma package (*Smyth, 2005*) from BioConductor (*Gentleman et al., 2004*). Log mean spot signals were taken for further analysis. Data were background corrected using the NormExp procedure on the negative control spots and quantile-normalized before averaging (*Smyth and Speed, 2003*). Genes were ranked for differential expression using a moderated *t*-statistic (*Smyth, 2004*). Results were uploaded to the National Center of Biotechnology Information Gene Expression Omnibus database (accession number GSE255669). List of differentially expressed genes is given in *Source data 1*.

## Proteomic analysis

Protein samples from isolated PASMC and PAAF were digested overnight at 37°C with Trypsin/LysC (Promega). Proteolytic digestion was quenched with 1% formic acid. The dried peptides were dissolved in reconstitution buffer (2% acetonitrile + 0.1% formic acid) and equimolar amounts (based on starting total protein concentration) of sample injected into the mass spectrometry instrument. Experiments were performed on the Orbitrap Fusion Tribrid mass spectrometer (Thermo Scientific) coupled with ACQUITY M-Class ultra-performance liquid chromatography (Waters Corporation). A flow rate of 450 nl/min was used for this liquid chromatography/mass spectrometry experiment, where mobile phase A was 0.2% formic acid in water and mobile phase B was 0.2% formic acid in acetonitrile. Analytical columns were pulled using fused silica (I.D. 100 µm) and packed with Magic 1.8 µm 120 Å UChrom C18 stationary phase (nanoLCMS Solutions) to a length of ~25 cm. Peptides were directly injected onto the analytical column using a gradient (2–45% B, followed by a high-B wash) of 80 min. The MS was operated in data-dependent fashion using CID (collision-induced dissociation) for generating MS/MS spectra, collected in the ion trap. Raw data were processed using Byonic v3.2.0 (ProteinMetrics) to infer protein isoforms using the Uniprot *Homo sapiens* database. Proteolysis with Trypsin/LysC was assumed to be semi-specific allowing for N-ragged cleavage with up to two missed cleavage sites. Precursor mass accuracies were held within 12 ppm and 0.4 Da for MS/MS fragments. Proteins were held to a false discovery rate of 1% or lower, using standard target-decoy approaches and only the proteins with >3 spectral counts were selected for further data processing (keratins and KAPs were removed at this stage). List of detected proteins is given in *Source data 2*.

## Gene/protein and pathway-centered functional data exploration and graphical representation

Gene set enrichment analysis (GSEA) and overrepresentation analysis of transcriptomics and proteomics data were performed by using gene ontology (GO) dataset (*Mi et al., 2019*) or BioPlanet (*Huang et al., 2019*) and applying the indicated significance and LFC cutoffs. The STRING (Version 11.0) Protein–Protein Interaction Networks/Functional Enrichment Analysis database was used to retrieve data on gene/protein interactions and examine the connections between genes. The gene-list enrichment tool Enrichr (*Chen et al., 2013*) was utilized to compare functional results. The open-source software platform Cytoscape (Version 3.7.2) was used for visualizing complex networks and representation of pathway terms with root nodes. Furthermore, RStudio (Version 1.1.383) and the package for circular visualization (*Gu et al., 2014*) and ggplot package (*Wickham, 2016*) were used for creating the circular plots.

## Multiplex immunofluorescent, RNA in situ hybridization and TUNEL staining

Formalin-fixed, paraffin-embedded lung sections were deparaffinized, rehydrated, and subjected to heat induced antigen retrieval. Multiplex immunofluorescence staining was done using Opal kit (Akoya). Successive rounds of primary antibody, detection reagent (Opal Polymer HRP, Akoya or Immpress HRP Polymer, Vector Labs), and fluorescent signal development (Opal dyes, Akoya or CF tyramide dye, Biotium), followed by antibody removal were performed according to the manufacturer's instructions. List of antibodies is given in *Supplementary file 2*. 4',6-Diamidino-2-phenylindol (DAPI) (Thermo Scientific) was used as a nuclear counterstain at 2.5 µg/ml final concentration. In

the case of single-molecule RNA in situ hybridization, slides were first against anti-PDGFRA antibody and processed further for RNA in situ hybridization according to the manufacturer's instructions using RNAscope Multiplex Fluorescent V2 assay (ACD). Hybridization probes against human IGFBP5, SCARA5 and CFD were obtained from ACD. Similarly, upon PDGFRA detection and proteinase K treatment, DNA cleavage in apoptotic cells was detected using the terminal deoxynucleotidyl transferase (TdT)-mediated dUTP nick-end labeling (TUNEL) method according to the manufacturer's instructions (Biotium). Following hybridization and TUNEL signal detection, slides were incubated with primary labeled antibody against alpha smooth muscle actin and counterstained with DAPI. Slides were imaged using SP8 fluorescence confocal microscope (Leica) and apochromatic glycerol immersion objectives 40× (1.25 numerical aperture).

## SeaHorse metabolic flux measurements

Isolated PASMC and PAAF (20,000/well) were plated directly into Seahorse XF96 cell culture microplates (Agilent Technologies) and allowed to reach confluency. After 96 and 120 hr the medium was replaced by Standard Oxidation Stress Test assay medium (10 mM glucose, 1 mM pyruvate, and 2 mM glutamine) and plates were incubated in a non-$CO_2$ incubator for 60 min prior to the assay. Oxygen consumption rate (OCR) and extracellular acidification rate (ECAR) were measured in a Seahorse XFe96 extracellular flux analyzer (Agilent) using XF Substrate Oxidation Stress Test Kits (XF Long Chain Fatty Acid Oxidation Stress Test Kit, XF Glucose/Pyruvate Oxidation Stress Test Kit). First, basal respiration was established, followed by injection of the relevant pathway inhibitor (4 µM etoxomir, 2 µM UK5099). The response to the inhibitor was monitored, then ATP production was blocked by oligomycin (1.5 µM), followed by the dissipation of proton gradients with FCCP (1 µM). The electron transport chain was inhibited by rotenone/antimycin (0.5 µM) to reveal the non-mitochondrial respiration. OCR/ECAR values are representing the ratio of mitochondrial respiration/glycolysis derived from the first rate measurements of the Substrate Oxidation Stress Test. The acute response to the inhibitor represents alterations of basal respiration as OCR response to pathway inhibitors (etoxomir, UK5099). The glycolytic capacity was calculated as the difference of maximum ECAR measurement after oligomycin injection and the last ECAR measurement right before oligomycin injection. Non-mitochondrial respiration was defined as the minimum OCR measurement after rotenone/antimycin injection. Proton leak was calculated as minimum OCR measurement after oligomycin injection minus non-mitochondrial respiration. Measurements were performed as septuplets, rate measurement values from each well were normalized to their corresponding protein amounts. Data were analyzed using the Wave Software (Agilent Technologies, Version 2.6.1).

## Proteoglycan/glycosaminoglycan quantification

Glycosaminoglycans were quantified in isolated cryo-preserved PA of donor and IPAH patients as described previously (*Dey et al., 1992*). Briefly, isolated PA were weighed and digested overnight at 60°C in a phosphate-EDTA buffer papain solution. Resulting lysate was mixed with 1,9-dimethylmethylene blue assay and the absorbance of the resulting complex measured spectrophotometrically at 525 nm. Chondroitin sulfate was used for standard curve preparation. Combination of Alcian blue and Verhoeff's staining of formalin-fixed, paraffin-embedded lung sections was used for localization of glycosaminoglycans and elastic fibers, respectively. Quantitative image analysis of Alcian blue staining was performed on digitalized tissue sections (VS200 Research Slide Scanner, Olympus Life Science) in QuPath software (https://qupath.github.io/, Version 0.5.1, *Bankhead et al., 2024*; *Bankhead et al., 2017*). Each identifiable pulmonary artery on the slide was annotated into adventital, medial and neointimal region of interest and for each region a median optical density (OD) measured using 'add intensity features' function. Absolute OD value for each region of each vessel is expressed relative to mean OD calculated from the donor medial regions.

## Cellular turnover measurements

Proliferation of PASMC and PAAF was determined by [³H]thymidine incorporation assay (BIOTREND Chemikalien, Germany). 7500 cells per well were seeded in a 96-well plate. The cells were starved in basal medium (without supplement) overnight and next day they were stimulated with 2% Fetal Calf Serum (FCS) or maintained in media without supplements as controls. Following 24 hr, [³H]thymidine incorporation measurement was evaluated in each well. Alternatively, cell turnover in different

passages was done by plating 100,000 cells in a flask and allowed to grow for 72 hr. Cells were detached by trypsination and cell number were determined using Neubauer cell counting chamber.

## Mitochondrial content, membrane potential, and cellular reactive oxygen species

Cellular reactive oxygen species (ROS) production was based on fluorescent imaging of the ROS indicator CellRox Deep Red (Thermo Scientific). Cells were loaded with 5 µM of CellROX dye for 1 hr at 37°C, followed by confocal fluorescence imaging (LSM 510 Meta with a Plan Neofluar 40×/1.3 oil-immersion objective, Carl Zeiss) before and after stimulation with 50 ng/ml PDGF-BB (R&D Systems) for 40 min at 37°C. Mitochondrial membrane potential was measured in quench/de-quench mode following staining with 1 µM tetramethylrhodamine methyl ester (TMRM) dye (Sigma-Aldrich). Image analysis was performed in ImageJ (NIH). A binary image of the TMRM signal was created, to measure the cell area covered by mitochondria. Mitochondrial density was then determined by dividing the cell area with the area covered with mitochondria. Mitochondrial membrane potential was calculated by normalizing the TMRM signal to the mitochondrial density.

## RNA isolation and quantitative real-time polymerase chain reaction

Total RNA from PAAF and PASMC was extracted using peqGOLD Total RNA Kit (Peqlab) or innuPREP RNA Mini Kit (IST Innuscreen). The cDNA was reverse transcribed using a qScript cDNA Synthesis kit (QuantaBio). Quantitative real-time polymerase chain reaction (qRT-PCR) was performed using the S'Green qPCR kit (Biozym) on LightCycler 480 (Roche Applied Science). The threshold cycle difference ($\Delta CT$) was calculated using the equation $\Delta CT = ([CT(B2M) + CT(HMBS)/2] - CT(\text{Gene of Interest})]$. Beta-2-microglobulin (B2M) and hydroxymethylbilane synthase (HMBS) were used as reference genes. The primer list is given in *Supplementary file 3*.

## Adhesion and gap closure assays

Adhesion assay was performed on Biocoat collagen I- and laminin-coated 24-well plates (Corning). Briefly, 50,000 cells were allowed to attach for 40 min at 37°C, followed by PBS wash, formalin fixation, and staining with crystal violet. Images were transformed into 8-bit image type and the adhered cells were counted with the plugin in cell counter of the Fiji-Image J. Gap closure assay was perfomed in two chamber silicon inserts with a defined cell-free gap were used (Ibidi). Each chamber seeded with 15,000 cells that were allowed to grow till confluency. Experiment was started by removing the insert and image of the cell-free gap taken ($t = 0$). Subsequent images of the gap were taken 4, 8, 24, and 48 hr after removal of the insert. Analysis of wound area has been performed with Fiji-ImageJ. Briefly, the wound area of each picture was first identified with the MRI Wound healing tool and the area enclosed was then measured.

## Cell stimulations and co-culture experiment

PASMC was cultured in 24- or 6-well plates up to 90% confluence and serum starved for 24 hr. Cells were stimulated for 24 hr with ligands C3a (R&D Systems, 100 ng/ml), C5a (R&D Systems, 100 ng/ml), pentraxin-3 (PTX3; R&D Systems, 5 µg/ml), hepatocyte growth factor (HGF; Peprotech, 25 ng/ml), chitinase 3-like 1 protein (R&D Systems, 300 ng/ml), insulin-like growth factor-binding protein-3 (R&D Systems, 300 ng/ml), and angiogenin (Biomol, 300 ng/ml) in starvation medium. Following stimulation, cells were washed with PBS and lysed for subsequent RNA isolation. For PAAF–PASMC co-culture experiment, cells were fluorescently labeled with either CellTracker Green CMFDA (Thermo Scientific) or CellTracker Red CMTPX (Thermo Scientific) for 30 min at 37°C. Labeled cells were trypsized and plated in a 1:1 ratio (PAAF:PASMC) in a 6-well plate in fully supplemented smooth muscle cell medium for 24 hr. Next day, cells were detached and single-cell suspension processed for two-way fluorescence-activated cell sorting. CMFDA and CMTPX viable cells (Sytox Blue dead cell stain negative cells) were sorted using FACSAria IIIu (BD Biosciences) at the flow cytometry core facility (Center for Medical Research, Medical University of Graz). Collected cells were centrifuged and processed for RNA isolation. One part of RNA was used for qRT-PCR, while other RNA was used for bulk RNA sequencing analysis. Briefly, libraries were prepared using NEB Next Ultra II RNA Library Prep Kit for Illumina (New England Biolabs) with rRNA Depletion kit v2 (Human/Mouse/Rat) (New England Biolabs) and sequenced upon library pooling on NovaSeq 6000 (Illumina). Results were uploaded to

the National Center of Biotechnology Information Gene Expression Omnibus database (accession number GSE262069). List of differentially expressed genes is given in *Source data 4*.

## Cell–cell communication analysis

Ligand–receptor expression analysis was performed on previously published single-cell RNA sequencing dataset of human pulmonary arteries from donors and PAH patients GSE210248 (*Crnkovic et al., 2022*). For the analysis, the data matrix was transformed into a Seurat object and was processed with the Seurat package (Version 4.3.0) (*Butler et al., 2018*) in the R studio environment (Version 4.1.2). The 10x files were converted into a Seurat object with CreateSeuratobject. Doublets were filtered out with the scDblFinder tool. Furthermore, all cells with more than 5% mitochondrial DNA and cells with under 200 and over 2500 RNA counts were filtered out. Final quality control step included filtering out outliers defined as cells with an RNA-count outside the range of 5 MADs. Data normalization was performed with the NormalizeData function from Seurat, followed by data scaling (ScaleData). The FindVariableFeatures Seurat-function detected the variable features in the dataset (selection method = vst, nfeatures = 2000). A principal component analysis was run on the normalized and scaled data. The number of PC was set after plotting an elbow plot which showed an elbow between 20 and 30 clusters. The following steps included data integration with harmony (Version 0.1.1) and uniform manifold approximation (UMAP) depiction of the data (RunUMAP). The data were clustered with the help of the FindNeighbours- and FindClusters function. The cluster annotation to cell type was done manually based on gene enrichment of canonical cell-type marker genes.

For the cell–cell communication analysis, the clustered datasets were normalized so that the number of cells per cell type in each group (PAH or donor) was comparable. Two independent analysis methods were used. A comparative Nichenet analysis was performed according to the Nichnet method (https://github.com/saeyslab/nichenetr, Version 2.0.4, *Browaeys and Sang-aram, 2023*; *Browaeys et al., 2020*). Donor and PAH niches were defined and smooth muscle cells and fibroblasts were each set as sender and receiver cell type. The analysis was performed twice to obtain results for communication in both directions. CellChat analysis: A descriptive analysis of cell–cell communication for donor and PAH samples was performed according to the CellChat tool (https://github.com/jinworks/CellChat, Version 1.6.1, *Jin, 2023*; *Jin et al., 2025*). The Seurat object was subset, and the analysis was performed separately for each subset.

## Secretome analysis

Proteome Profiler Human XL Cytokine Array Kit (R&D Systems) and Proteome Profiler Human Adipokine Array Kit (R&D Sytems) were used to measure factors secreted by donor and IPAH-PAAF. Briefly, PAAF were grown till 80–90% confluence and after 3–4 days medium was collected, centrifuged at 4°C, 400 × *g* for 5 min, aliquoted and stored at –80°C till measurement. Equal volume of cell culture supernatant was used in Proteome Profiler Array Kits and assay performed according to the manufacturer's instructions. Signal intensity values for each spot were normalized to the mean value of the positive control spots for each membrane. Normalized intensity values were then used to calculate pairwise (IPAH-to-donor) relative expressions for each factor above the detection limit. In total, four such pairwise comparisons were done. We took into further consideration only those factors whose relative normalized expression levels changed by more than 10% in at least two pairwise comparisons. Obtained list of ligands with corresponding normalized relative expressions was hierarchically clustered using Clustergrammer online tool (*Fernandez et al., 2017*) to provide a heatmap and correlation matrix visualization.

## Statistical analysis

Graphs and statistical calculations were performed using GraphPad Prism (Version 8.0, GraphPad Software) or R (v3.5.3, packages stringr, data.table, readxl, openxlsx, MetaboAnalystR, ggplot2, colorspace, circlize) and Tibco Spotfire (v11.1.0). Probability values of p < 0.05 were considered statistically significant. Differences between groups were investigated using Student's *t*-test, Mann–Whitney, or Friedman test. Two- or three-way ANOVA was used for interaction analysis. For orthogonal partial least squares discriminant analysis (OPLS-DA) data were filtered to retain only features with less than 30% missing data (proteomics 1552 genes, transcriptomics 23,541 genes). OPLS-DA was performed centered and scaled to unit variance with a standard sevenfold cross validation for the classification

factor. Model stability was additionally verified with 1000 random label permutations and models with Q2 >50% were considered significant.

## Results

### Distinct omics profiles underlie specialized homeostatic functions of normal PASMC and PAAF

The overarching aim of the current study was to characterize PASMC and PAAF functionality beyond their classical contractile and structural roles. At first, we focused on defining the normal, healthy cell states by using very early-passage (p1) PASMC and PAAF isolated from same (source-matched) pulmonary arteries (PA) of healthy, downsized donors ($n = 4$). We performed deep molecular characterization of RNA and protein expression profiles (*Figure 1A*, *Source data 1*, and *Source data 2*). Principle component analysis confirmed that molecular distinction between two cell types was preserved in early passage on both transcriptomic and proteomic level (*Figure 1B*). It also revealed higher interpersonal heterogeneity within PASMC samples, while source-matched PAAF represented as a more homogenous group (*Figure 1B*). We identified a set of significantly enriched genes/proteins in PASMC (Regulator of G Protein Signaling 5 – RGS5, Insulin-Like Growth Factor-Binding Protein 5 – IGFBP5, Integrin Alpha 2 – ITGA2, Cysteine and Glycine Rich Protein 1 – CSRP1, Actin Filament-Associated Protein 1 – AFAP1) and PAAF (Alcohol Dehydrogenase 1C – ADH1C, Scavenger Receptor Class A Member 5 – SCARA5, Complement Factor D – CFD, Microsomal Glutathione *S*-Transferase 1 – MGST1, Thioredoxin-Related Transmembrane Protein 2 – TMX2) for orthogonal validation (*Figure 1C*). Some of the molecules were previously associated predominantly with SMC, such as RGS5 and CSRP1 (*Crnkovic et al., 2022*; *Snider et al., 2008*), or adventitial fibroblast, such as SCARA5, CFD, and MGST1 (*Crnkovic et al., 2022*; *Sikkema et al., 2023*) expression. We mapped the location of selected enriched targets by multicolor staining of human lung tissue (*Figure 1D, E*: $n = 5$). In the pulmonary artery compartment, RGS5 and IGFBP5 positivity overlapped predominantly with ACTA2 (alpha smooth muscle actin) stain, while ADH1C, SCARA5, and CFD were observed mostly in adventitial region marked by PDGFRA (Platelet-Derived Growth Factor Receptor Alpha) (*Figure 1D, E*). Antibodies used to detect other targets (ITGA2, AFAP1, MGST1, and TMX2) yielded inconclusive results due to lack or very weak signal in PASMC/PAAF, with exception of CSRP1. AFAP1 and ITGA2 stained endothelial layer, CSPR1 had medial staining pattern in addition to inflammatory cells, while MGST1 and TMX2 stained mostly inflammatory or epithelial cells (*Figure 1—figure supplement 1*). Comparative analysis of transcriptomic and proteomic datasets revealed a strong, but not complete level of linear correlation between the gene and protein expression profiles (*Figure 1—figure supplement 1*). We therefore decided to use an integrated dataset and analyzed all significantly enriched genes and proteins ($-\log_{10}(p) > 1.3$) between both cell types to achieve stronger and more robust analysis (*Figure 1F*). A GSEA on GO terms for biological processes was used to infer functional implications of observed expression profiles. The analysis confirmed polarized and distinct cell-type functional associations, revealing vascular development and cell migration processes for PASMC, and endoplasmatic reticulum protein targeting for PAAF. Additionally, PASMC was associated with processes linked to sterol transport, *O*-glycosylation, and nerve growth factor signaling (*Figure 1F*). PAAF on the other hand showed enrichment in mitochondrial metabolic processes and heavy metal ion detoxification (*Figure 1F*). Applying an additional cutoff by taking into consideration only strongly expressed genes/proteins, we performed an overrepresentation analysis of GO biological processes (*Figure 1G*). Among top10 processes, there was a clear separation and functional distinction between PASMC and PAAF, with overlap in angiogenesis and adhesion (*Figure 1G*). Contrasting the canonical functional roles, it was PASMC, rather than PAAF, that showed enrichment for ECM organization (*Figure 1G*), while PAAF showed enrichment for heavy metal ion response, lipid homeostasis, and cell differentiation (*Figure 1G*). The enrichment of lipid-associated processes in PAAF prompted us to validate the observed results phenotypically. We performed a fuel usage assay using Seahorse metabolic analyzer ($n = 6$). Real-time monitoring of oxygen consumption under basal condition and upon specific metabolic inhibitors revealed that PASMC could equally utilize glucose/pyruvate and long chain fatty acids as a fuel source for oxidative metabolism (*Figure 1H1*). In contrast, upon blocking fatty acid utilization in Krebs cycle, PAAF had stronger drop in basal respiration rate while PASMC was less sensitive to this maneuver (*Figure 1I*). PAAF displayed additional significant metabolic differences

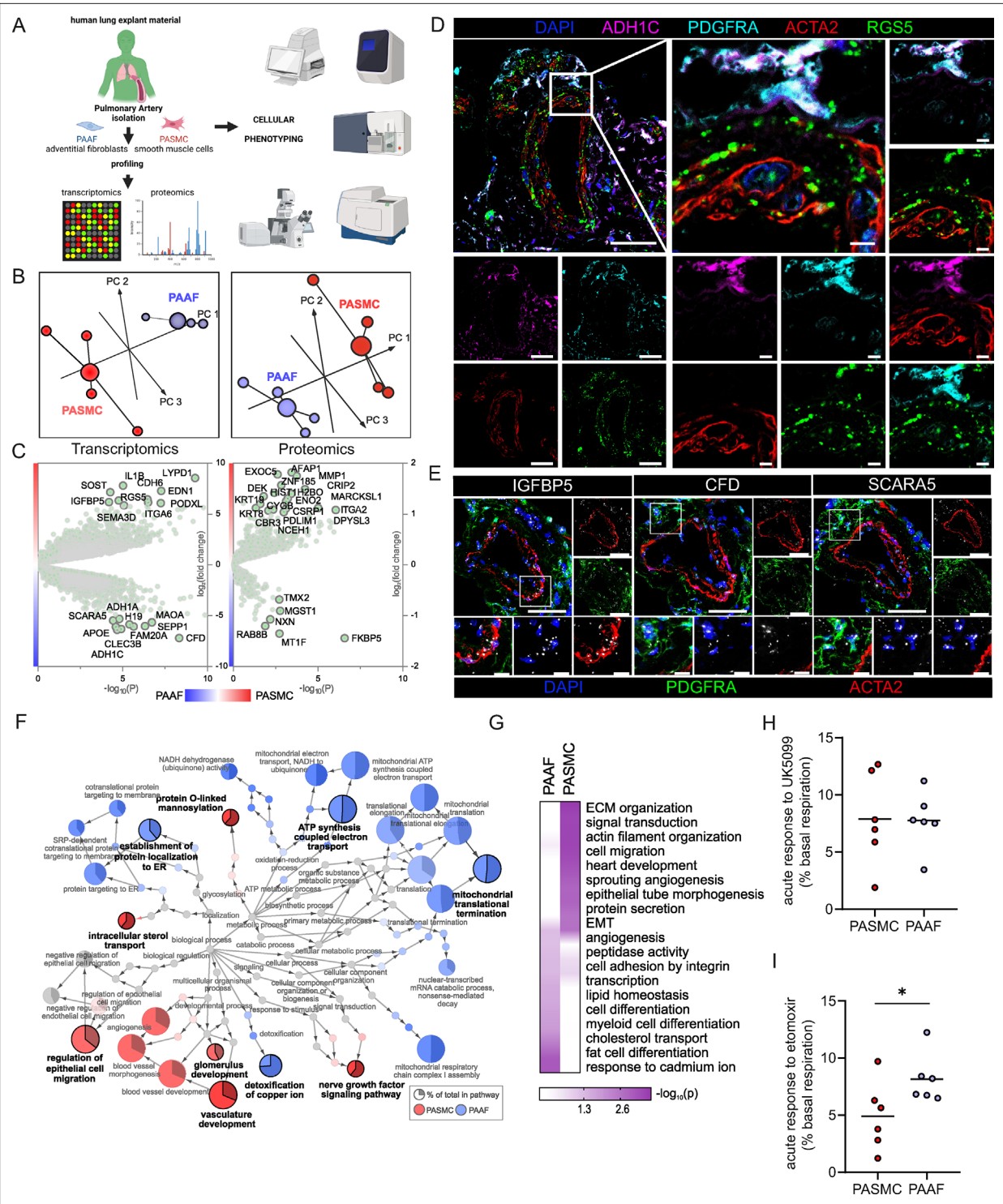

**Figure 1.** Omics-assisted phenotypic characterization of cell states in healthy human pulmonary artery smooth muscle (PASMC) and adventitial fibroblast (PAAF) lineages. (**A**) Schematic representation of the experimental setup using early-passage cells (*n* = 4). Created with BioRender.com. (**B**) 3D score plots of principal component analyses (PCA), larger nodes represent gravity centers. (**C**) Volcano plots of log₂ fold change between donor– PASMC and PAAF plotted against significance (−log₁₀(p)). Genes names depicted for the top 20 transcriptomics hits and proteins above the threshold (−log₁₀(p) >1.3 |LFC|>1). (**D**) Immunoflourescent localization of PASMC (ACTA2 and RGS5) and PAAF markers (PDGFRA and ADH1C) in normal human lungs (*n* = 5). (**E**) In situ hybridization localization of PASMC (IGFBP5) and PAAF markers (CFD and SCARA5). 4',6-Diamidino-2-phenylindol (DAPI) as nuclear counterstaining. White bar depicting 50 µm (5 µm for zoomed in panels). (**F**) Gene set enrichment analysis (GSEA) of all significantly regulated transcriptomics and proteomics targets between donor–PASMC and PAAF using the gene ontology (GO) database. Parent-to-root node visualization

*Figure 1 continued on next page*

*Figure 1 continued*

(intermediate terms omitted) with node size reflecting significance. Highlighted nodes depict significantly altered GO overview terms being higher expressed in either PASMC or PAAF. (**G**) Top GO terms resulting from an overrepresentation analysis (ORA) of the omics dataset using more stringent cutoff values ($-\log_{10}(p) >3$ |LFC|>2 transcriptomics; $-\log_{10}(p) >1.5$ |LFC|>0.5 proteomics). (**H**) Calculated change in basal oxygen consumption rate upon addition of UK5099 (glucose/pyruvate mitochondrial uptake inhibitor) and (**I**) etomoxir (long chain fatty acid mitochondrial uptake inhibitor). Mann–Whitney test, '*' denotes $p < 0.05$.

The online version of this article includes the following figure supplement(s) for figure 1:

**Figure supplement 1.** Validation and correlation between transcriptomics and proteomics.

**Figure supplement 2.** Real-time metabolic analysis between donor pulmonary artery smooth muscle cell (PASMC) and pulmonary artery adventitial fibroblast (PAAF).

to PASMC (*Figure 1—figure supplement 2*). PAAF had lower OCR and ECAR (*Figure 1—figure supplement 2*). Ratio of oxygen consumption to extracellular acidification (OCR/ECAR) revealed that PAAF has relative preference for oxidative phosphorylation compared to PASMC (*Figure 1—figure supplement 2*), in line with observed enrichment for mitochondrial respiration processes (*Figure 1F*). This was further supported by lower glycolytic capacity (*Figure 1—figure supplement 2*) and lower mitochondrial proton leak (*Figure 1—figure supplement 2*). Cumulatively, molecular and phenotypic profiling revealed specialized homeostatic roles of normal, healthy PASMC, and PAAF.

## Preservation of major PASMC- and PAAF-defining features and acquisition of novel IPAH cell states

We next set out to delineate expression changes defining the diseased, remodeling state. Omics profiling was performed on source-matched PASMC and PAAF isolated from patients with end-stage idiopathic pulmonary arterial hypertension (IPAH, *n* = 4) in comparison to donor samples (*Figure 1A*). PCA analysis of transcriptomic and proteomic profiles revealed a clear sample separation based on cell type (*Figure 2A*). However, sample separation based on disease state was less obvious (*Figure 2A*). We used the supervised multivariate OPLS-DA method to determine the significance of cell-type and disease-state as parameters that drive sample separation. In the first model iteration, we considered either cell type or disease state as input parameter. OPLS-DA modeling found significant differences in the global transcriptomics and proteomics profiles based only on the cell type, but not disease state (*Figure 2B*), validating PCA analysis observations. For the second model iteration, we took both cell type and disease state simultaneously as input parameters. In this OPLA-DA model, only in the case of transcriptomic dataset there was significant separation between all four groups (*Figure 2C*). These results imply that remodeling process results in substantial transcriptional changes (cell state), but that each cell type retains their original cell lineage designation, in line with results from fate mapping and fresh cell single-cell transcriptomics (*Biasin et al., 2020*; *Crnkovic et al., 2022*; *Crnkovic et al., 2018*). We accordingly evaluated the preservation of the expression profile and localization in IPAH of donor–PASMC- and PAAF-enriched targets. Multicolor staining of IPAH lungs revealed the preserved staining patterns in the pulmonary artery compartment compared to donors, with RGS5 signal associating with ACTA2, while ADH1C, SCARA5, and CFD signals associated with PDGFRA (*Figure 2D*, *Figure 2—figure supplement 1*). IGFBP5 in situ hybridization signal was detected both in PASMC and PAAF regions (*Figure 2D*). Comparable patterns of target expression were further corroborated by single-cell transcriptomics data from fresh human pulmonary arteries (*Crnkovic et al., 2022*) with the notable exception of IGFBP5, which was predominantly expressed in PAAF (*Figure 2E*).

## IPAH-defining cell states of PASMC and PAAF in vitro

We next focused on characterizing disease-state-defining features. GO analysis of selected gene sets from combined transcriptomic and proteomic data (significantly regulated genes/proteins), showed a cell-type-specific footprint of molecular changes upon IPAH (*Figure 3A*). IPAH-PASMC were characterized by upregulation of nucleotide-sugar and lipid biosynthesis, Golgi vesicle transport and endoplasmic reticulum (ER) stress processes, while simultaneously displaying downregulation of cytoskeletal components, ribonucleoprotein complex export and p53 signaling (*Figure 3A*). IPAH-PAAF upregulated genes and proteins involved in cell proliferation, DNA repair and downregulated expression of NF-κB pathway components, NK-cell activation, and transcriptional stress response

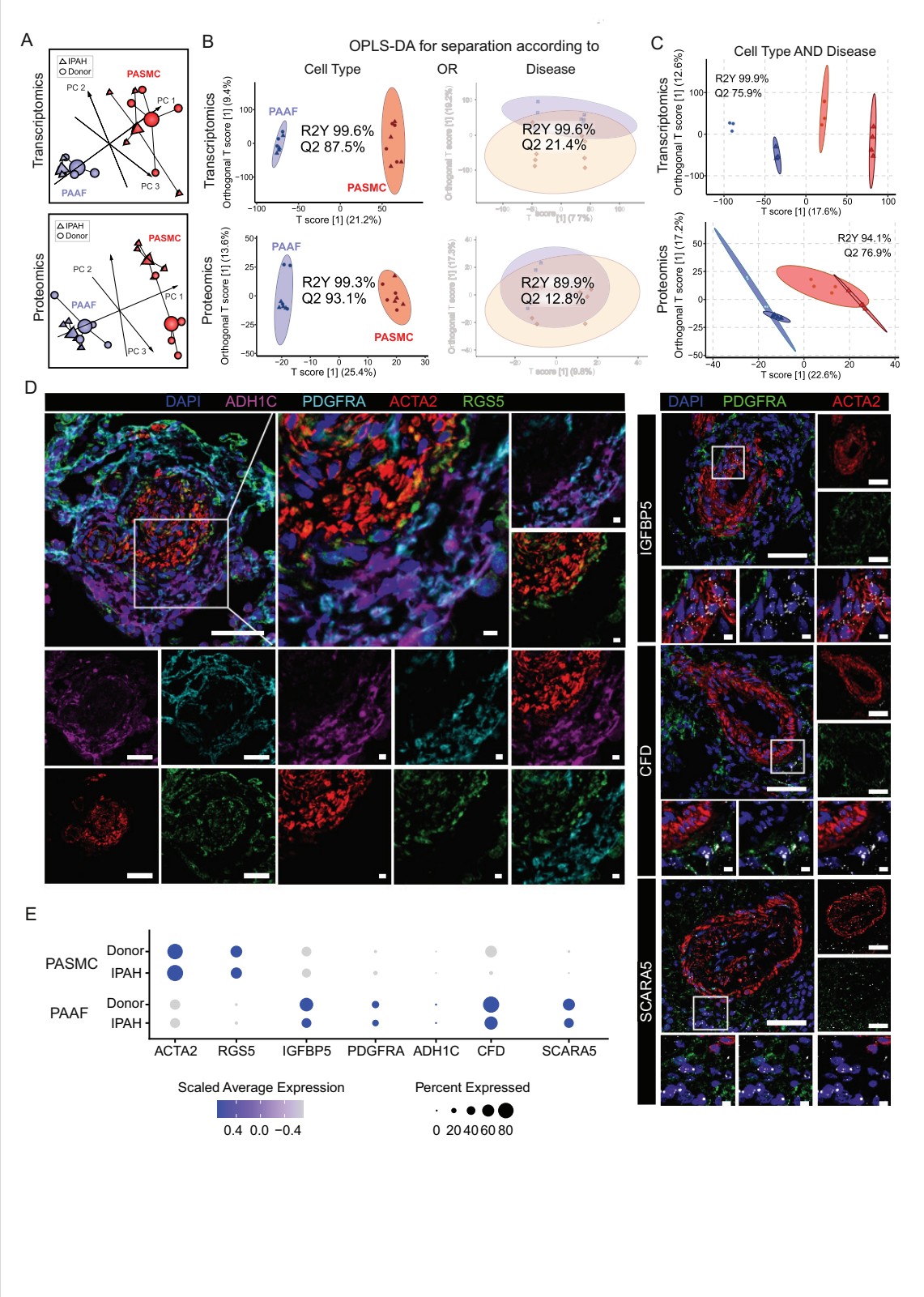

**Figure 2.** Preserved lineages and distinct pulmonary artery smooth muscle cell/pulmonary artery adventitial fibroblast (PASMC/PAAF) cell states in idiopathic pulmonary arterial hypertension (IPAH). (**A**) 3D score plots of principal component analyses (PCA), larger nodes represent gravity centers. (**B**) Orthogonal projection to latent structures-discriminant analysis (OPLS-DA) *T* score plots separating predictive variability (*x*-axis), attributed to biological grouping, and non-predictive variability (technical/inter-individual, *y*-axis). Monofactorial OPLS-DA model for separation according to cell type or

*Figure 2 continued on next page*

*Figure 2 continued*

disease. (**C**) Bifactorial OPLS-DA model considering cell type and disease simultaneously. Ellipse depicting the 95% confidence region, Q2 denoting model's predictive power (significance: Q2 >50%) and R2Y representing proportion of variance in the response variable explained by the model (higher values indicating better fit). (**D**) Immunoflourescent and in situ hybridization localization of PASMC (ACTA2, RGS5, and IGFBP5) and PAAF markers (PDGFRA, ADH1C, CFD, and SCARA5) in IPAH human lungs (*n* = 5). White bar depicting 50 μm (5 μm for zoomed in panels). (**E**) Dot plot showing relative expression of omics-identified PASMC/PAAF markers in published single-cell transcriptomics dataset of fresh human pulmonary arteries of donors and IPAH patients (GSE210248).

The online version of this article includes the following figure supplement(s) for figure 2:

**Figure supplement 1.** Validation of omics markers in idiopathic pulmonary arterial hypertension (IPAH).

(*Figure 3A*). Enrichment for activated sugars and Golgi transport process is associated with increased synthesis and secretion of glycosylated proteins, such as ECM components. We therefore measured production of proteoglycans on isolated PA using 1,9-dimethylmethylene blue assay. Quantitative analysis revealed increased GAG content in IPAH (*n* = 8) compared to donor (*n* = 7) PA (*Figure 3B*). We localized the site of the increased GAG content in PA using Alcian blue staining of human lung tissue. This showed enhanced medial staining in IPAH (*n* = 10) compared to donor (*n* = 10) and prominent signal in neointimal PA compartment (*Figure 3C*) that was confirmed by quantitative image analysis (*Figure 3D*). In line with observed enrichment for DNA replication and cell cycle phase regulation (*Figure 3A*), IPAH-PAAF displayed higher proliferative capacity compared to donor–PAAF, upon both stimulated and unstimulated conditions (*Figure 3E*). IPAH-PASMC in contrary showed lower proliferative rate compared to donor–PASMC (*Figure 3E*), corresponding to our previous observation of G1 cell cycle accumulation of IPAH-PASMC (*Crnkovic et al., 2022*). The observed phenotypic effect is strongest in very early passage and is progressively lost with passaging (*Figure 3F*). In vitro results mirror indirect determination of cellular proliferation rates in situ using PCNA staining of human lung tissue. We observed frequent PCNA+ fibroblasts, identified by PDGFRA staining, in the adventitial layer of PA from IPAH patients (*Figure 3G*), while decreased percentage of proliferating PASMC in IPAH PA, detected by PCNA and Ki-67 staining, was reported previously (*Crnkovic et al., 2022*). The attenuated differences in proliferation rates between IPAH and donor cells with prolonged passaging suggest culture-induced cell-state changes and a loss of disease-specific phenotypic features. To investigate the stability of key phenotypic differences between IPAH and healthy cells, we analyzed transcriptional signatures in freshly isolated, early-passage, and late-passage cells using published datasets (*Chelladurai et al., 2022*; *Crnkovic et al., 2022*; *Gorr et al., 2020*). GSEA revealed significant distinctions between fresh and cultured cells, with marked differences in the top-enriched biological processes across all conditions (*Source data 3*). By focusing on the key phenotypic features identified in this study (*Figure 3A*), we found that IPAH PASMCs exhibit a more gradual phenotypic shift with passaging, in contrast to the sharp changes observed early on in cultured PAAFs (*Figure 3H*). Early-passage IPAH-PASMC (p1) retain positive enrichment for biosynthetic and ECM processes and share positive enrichment for macrophage regulation processes with fresh cells, while simultaneously displaying gradual negative enrichment of contraction associated processes (*Figure 3H*). Surprisingly, both early (p1)- and late-passage (p6) IPAH-PAAF display prominent positive enrichment for cell cycle and proliferation processes, with fresh cells showing weak enrichment for DNA recombination process (*Figure 3H*). Even more striking are prominent inverse enrichments for inflammatory/NF-κB signaling between fresh and cultured IPAH-PAAF (*Figure 3H*). Similar, but less pronounced trend for inverse enrichment is displayed for processes involving cellular adhesion and ECM organization (*Figure 3H*). In summary, expression profile of early-passage cells retains some of the key phenotypic features displayed by cells in their native environment.

## Shared IPAH expression disturbances in PASMC and PAAF converge on mitochondrial dysfunction

Distinct remodeling expression profiles and cell behavior, led us to search if there are any common molecular perturbations shared between IPAH-PASMC and IPAH-PAAF. Number of co-directionally regulated genes and proteins were limited to 47 (*Figure 4A*). Further selection based on significance and fold change provided a list of most significantly regulated molecules in IPAH that are shared between PASMC and PAAF (*Figure 4B*, highlighted corners bottom left and top right). Using this as an input for network analysis based on the STRING database, we built the underlying gene interaction

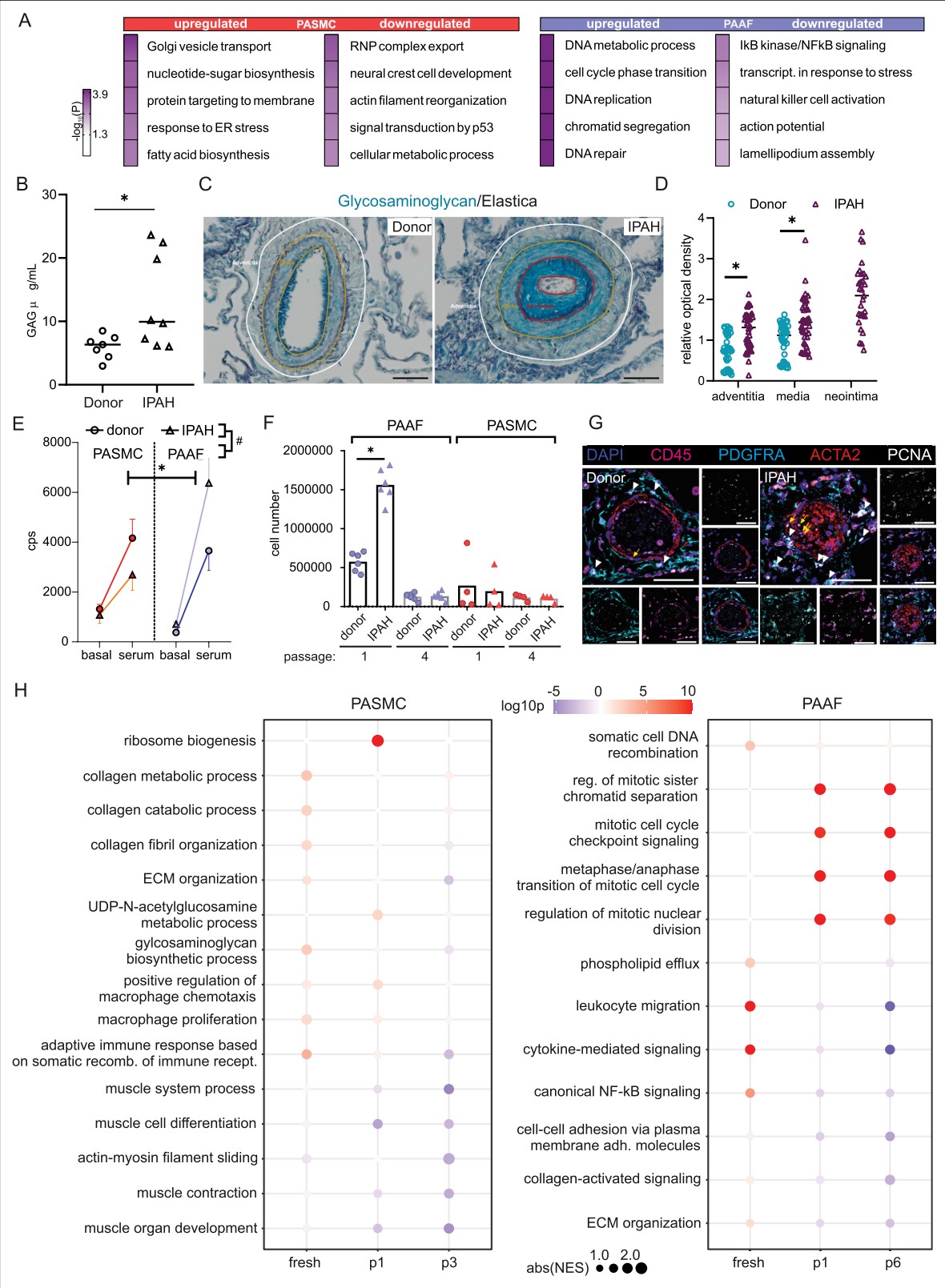

**Figure 3.** Phenotypic correlates of pulmonary artery smooth muscle cell/pulmonary artery adventitial fibroblast (PASMC/PAAF) cell state in idiopathic pulmonary arterial hypertension (IPAH). (**A**) Top 5 biological process terms from the gene ontology enrichment analysis that are up- or downregulated in IPAH. (**B**) Dimethylmethylene blue (DMMB) assay for quantification of glycosaminoglycan content in isolated pulmonary arteries from donor (*n* = 7) or IPAH (*n* = 8) patients. Mann–Whitney test. (**C**) Representative Alcian blue (glycosaminoglycans – blue) with Verhoeff's staining (elastic fibers – black/

*Figure 3 continued on next page*

Figure 3 continued

gray) of donor and IPAH lungs. Scale bar: 100 µm. Annotated regions depicting adventitia (white), media (yellow), and neointima (red). (D) Quantitative image analysis of Alcian blue staining intensities in adventitial, medial, and neointimal regions of pulmonary arteries from donors (n = 10) and IPAH patients (n = 10). Two-way ANOVA followed by Sidak's multiple comparisons test (*p < 0.05). (E) Proliferative response of passage 3 cells measured by [$^3$H]-thymidine incorporation assay upon serum stimulation. Dots represent mean values (n = 5–6 donors/IPAH) with bars showing standard errors of mean. Interaction effects of stimulation, cell type, and disease state on cellular proliferation were analyzed by three-way ANOVA. Significant interaction effects are indicated as follows: * for stimulation × cell type interactions and # for cell type × disease state interactions (both *, #p < 0.05). (F) Absolute cell counts measured after 24 hr growth. Mann–Whitney test, p < 0.05. (G) Representative immunofluorescent localization of proliferating (PCNA marker) PASMC (ACTA2, yellow arrow) and PAAF (PDGFRA, white arrowhead) in health (donor) and diseased (IPAH) lungs. Immune cells were identified through CD45 expression. 4′,6-Diamidino-2-phenylindol (DAPI) nuclear counterstain, 50 µm scale bar. (H) Comparison of gene expression changes in IPAH cells compared to healthy state in three conditions (fresh – GSE210248; passage 1 – GSE255669; and passage 3 – GSE144274 or passage 6 – GSE144932). Dot plot of gene set enrichment analysis for selected biological process terms with dot size depicting absolute value of the normalized enrichment score (absNES), color intensity showing significance level (log$_{10}$p) and color coding for up- or downregulation of the biological process.

network and identify shared upstream regulating factors (*Figure 4C*). Only a fraction of commonly regulated genes and proteins were encompassed by the putative regulatory network (*Figure 4C*). For the remaining molecules, it was not possible to construct an interaction map, implying regulation by independent mechanisms and upstream regulators. We therefore focused further on those molecules that were encompassed in the interaction network. Interestingly, their regulatory nodes converged on the immuno-inflammatory and survival-apoptosis regulators and effectors (*Figure 4C*). Specifically, a majority of the nodes building the interaction network belonged to a limited set of genes comprising mostly cytokines/chemokines (IL1B, IL2, IL4, IL6, IL10, TNF, CCL2, and CXCL8) and inflammatory-stress response signal transduction pathways (JNK, AKT, NF-κB, STAT3, MYC, and JUN) (*Figure 4C*). Additional pathway enrichment analysis based on the BioPlanet database, uncovered regulation of apoptosis as a shared pathway by both cell types in IPAH (*Figure 4D*). Shared SOD2 (mitochondrial superoxide dismutase 2) downregulation prompted us to investigate the mitochondrial function and redox defenses. We performed TMRM staining for the detection of mitochondrial content and potential. Cellular mitochondrial content was lower both in IPAH-PASMC and IPAH-PAAF upon removal of serum and growth factors (starvation) (*Figure 4E*). However, the remaining mitochondria in IPAH cells were hyperpolarized (*Figure 4F*). Mitochondrial membrane hyperpolarization is linked with increased ROS production. We measured cellular ROS production using CellRox Deep Red dye at basal levels and upon PDGF-BB stimulation revealing that only IPAH-PASMC significantly upregulate ROS production (*Figure 4G*). Although mitochondrial dysfunction seems as a common underlying disturbance between cell types in IPAH, the downstream consequences appear to be cell-type selective. Mitochondrial dysfunction and elevated ROS production are associated with increased apoptosis, aligning with reduced SOD2, which plays a role in clearing mitochondrial ROS and mitigating oxidative stress to protect against cell death (*Kokoszka et al., 2001*; *Figure 4D*). We therefore assessed in situ the rate of apoptosis using TUNEL assay on human lung tissue samples from donors and IPAH patients (n = 6 for each condition) and co-stained with ACTA2 and PDGFRA (*Figure 4H*). However, apoptosis measured as TUNEL+ cells was in both cases a very rare event (1–2 positive ACTA2 or PDGFRA PA cells per slide) that no meaningful quantification was possible.

## PASMC and PAAF acquire divergent disease-cell states

Contrasting the limited overlap in regulated elements between PASMC and PAAF, vast majority of IPAH transcriptomic and proteomic changes (more than a thousand genes/proteins) were uniquely regulated between the cell types, including substantial number of inversely regulated molecules (*Figure 5A*). PASMC possessed more pronounced changes than PAAF as shown by the number of differentially regulated genes and proteins in IPAH (*Figure 5B*). A STRING-based interaction network, overlaid with information about the initial cell-type association under normal condition (preferential enrichment in donor–PASMC or PAAF) (*Figure 5B*), revealed that the majority of downregulated molecules in IPAH-PASMC came from a set of genes and proteins preferentially expressed by PASMC in healthy state (donor–PASMC) (*Figure 5B*). This reflects the fact that most downregulation in PASMC takes place in contractile machinery elements (*Figure 3A*) that are characteristic for smooth muscle cell lineage. In contrast, molecules upregulated in IPAH-PASMC contained targets from both PASMC- and PAAF-enriched dataset under healthy cell states (donors). IPAH-regulated molecules in PAAF on the other hand could be traced back to both PASMC- and PAAF-enriched normal cell state

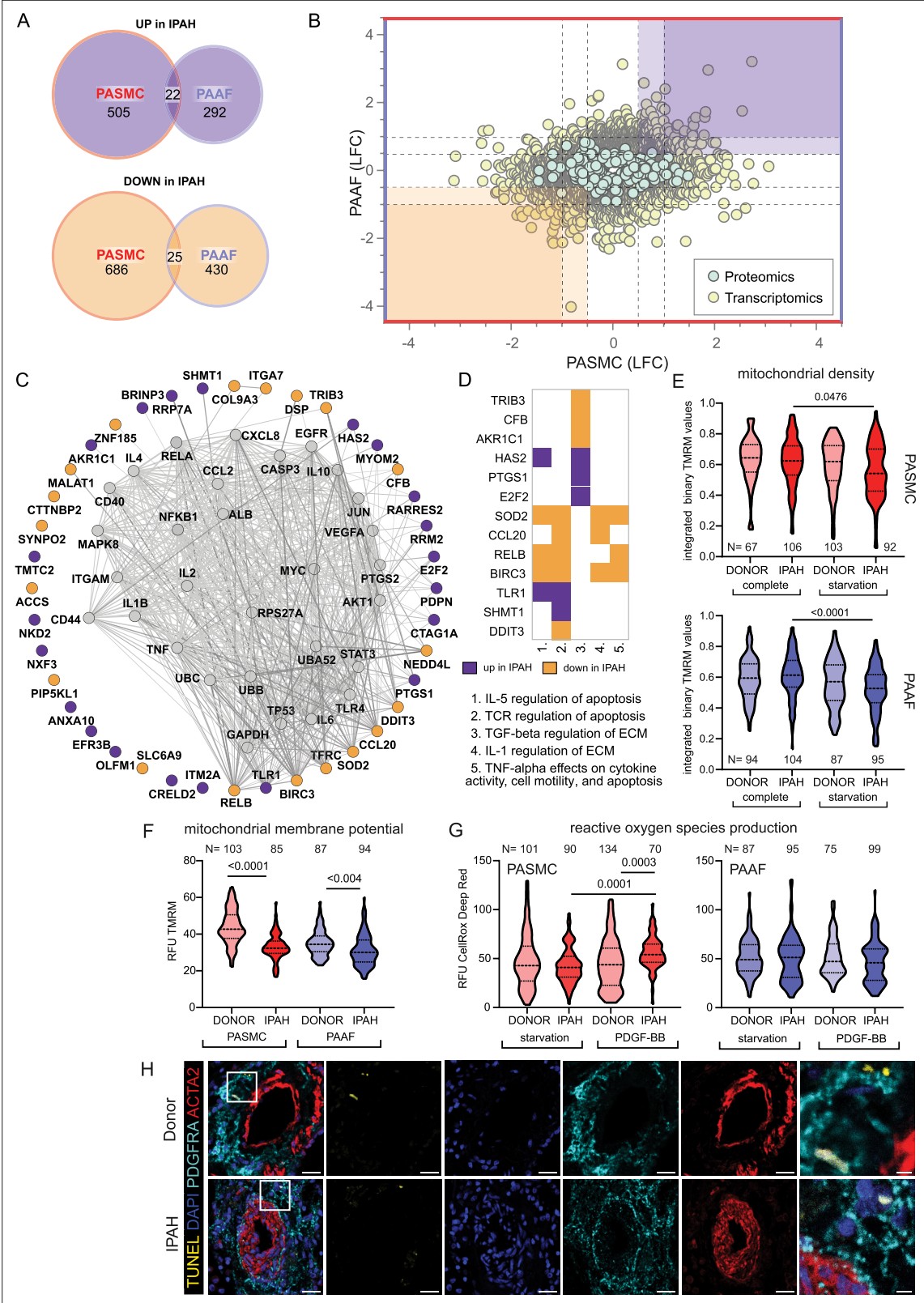

**Figure 4.** Mitochondrial dysfunction as an intersecting phenotypic characteristic of pulmonary artery smooth muscle cell (PASMC) and pulmonary artery adventitial fibroblast (PAAF) in idiopathic pulmonary arterial hypertension (IPAH). (**A**) Euler diagram of the differentially expressed genes summarizing the overlaps and disjoints in regulation (criteria employed: $-\log_{10}(p) > 1.3$; and LFC ±|1| for transcriptomics data and ±|0.5| for proteomics data). (**B**) Scatter plot graph of $\log_2$ fold changes between donor and IPAH in PASMC plotted against changes in PAAF, shaded areas highlighting commonly

*Figure 4 continued on next page*

*Figure 4 continued*

regulated genes/proteins. (**C**) Visualization of STRING-based interaction and regulatory network with gray nodes representing identified putative upstream regulators. (**D**) Pathway analysis performed in Enrichr using the BioPlanet database with a matrix annotating the main genes involved with color coding reflecting the IPAH dependent regulation. (**E**) Mitochondrial content measurement using tetramethylrhodamine methyl ester (TMRM) dye in complete or starvation (no serum) medium. (**F**) TMRM dye (quench mode)-based mitochondrial membrane potential measurement following starvation. (**G**) Basal and PDGF-BB-stimulated reactive oxygen species production (ROS) (40 min, 50 ng/ml) using CellRox DeepRed dye. Kruskal–Wallis test followed by Dunn's multiple comparisons test. Cell numbers from each condition and cell type denoted by N on the graph (n = 3 independent donors/IPAH) (**E–G**). (**H**) Representative immunofluorescent localization of apoptotic cells (TUNEL marker). PASMC (ACTA2 marker) and PAAF (PDGFRA marker) in health (donor) and diseased (IPAH) lungs (n = 6). 4′,6-Diamidino-2-phenylindol (DAPI) nuclear counterstain, 25 μm scale bar (5 μm for zoomed in panel).

(*Figure 5B*). GO analysis of IPAH revealed a clear cell-type distinct fingerprint of molecular changes: PASMC harbored expression changes in biological processes related to ECM, while cell cycle and replication processes were enriched in PAAF (*Figure 5C*). Looking selectively at top 20 differentially regulated genes, a similar pattern was observed: genes that were downregulated in IPAH-PASMC, such as SHROOM3, BLID or CCK, were preferentially expressed by donor–PASMC, while upregulated genes such as chemokines (CCL7 and CCL26) were not restricted to smooth muscles (*Figure 5D*). For comparison, expression of the same genes in PAAF was not significant (*Figure 5D*). The same was observed for top 20 differentially regulated genes in IPAH-PAAF – their expression was not significantly changed in PASMC (*Figure 5E*). However, osteoprotegerin (TNFRSF11B) and matrix metalloproteinase 1 (MMP1), genes with most pronounced expression change in IPAH-PAAF, are enriched in PASMC (*Figure 5E*). These examples show the importance of distinguishing cell-type-specific contributions to gene/protein expression levels. It also implies cell-type-specific regulatory mechanisms at play.

Biological function assignment of top 20 regulated elements in PASMC and PAAF entailed a broad spectrum of assigned biological functions, ranging from motility, adhesion, hydrolysis, metabolism, growth, differentiation, and ECM components. We wondered if a gene/protein-level evaluation of significantly changed elements for each cell type between IPAH and donors, could give indication of putative phenotypic changes in addition to pathway enrichment approach. We manually curated significantly changed elements in IPAH-PASMC (*Figure 5F*, outermost circle) and IPAH-PAAF (*Figure 5F*, middle circle) and grouped them in ECM, immune system, metabolism, and cell cycle categories (*Figure 5F*). Elements that additionally had significantly differing regulations between PASMC and PAAF under IPAH conditions were marked bold (*Figure 5F*, black bordering and bold font). We also provide reference if a particular element was preferentially expressed in PASMC or PAAF (*Figure 5F*, innermost circle). Striking feature is that both cell types display perturbations in all four major functional categories, the involved genes/proteins are different. For instance, IPAH-PASMC upregulate proinflammatory cytokines (CCL7 and CCL26) and ECM components claudin 14 (CLDN14) and metalloproteinase ADAMTS18, while IPAH-PAAF downregulate anti-inflammatory NF-κB component RelB and laminin 1 (LAMA1) and matrix metallopeptidase 1 (MMP1) ECM components (*Figure 5F*). Generally, PASMC and PAAF take a divergent path upon vascular remodeling: while IPAH-PASMC acquire synthetic phenotype with up regulation of chemotactic elements, IPAH-PAAF display hallmarks of fast-cycling cells. Based on these results, we wondered if ECM-based phenotypic assays could uncover cell-type and cell-state distinct responses, similarly to results obtained by proliferation and metabolic assays.

## ECM components centered signaling elicits cell-type and cell-state distinct phenotypic responses

Changes in ECM composition are a recurring hallmark of pulmonary vascular remodeling (*Hoffmann et al., 2015*; *Hoffmann et al., 2014*; *Jandl et al., 2020*; *Mutgan et al., 2024*) recapitulated also in current omics dataset (*Figure 5F*). Ligand–receptor interaction analysis of our omics dataset indicated that ligand expressing PAAF might act as sender cell type and PASMC as receiver cells through their integrin receptor repertoire expression (*Figure 6A*). ITGA2 and ITGB1 subunits expressed in PASMC build a heterodimer complex functioning both as laminin and collagen receptor (*Languino et al., 1989*; *Figure 6B*). We speculated that change in laminin-to-collagen ratio in ECM would have functional consequence and influence phenotypic behavior of cells in contact with altered ECM. We

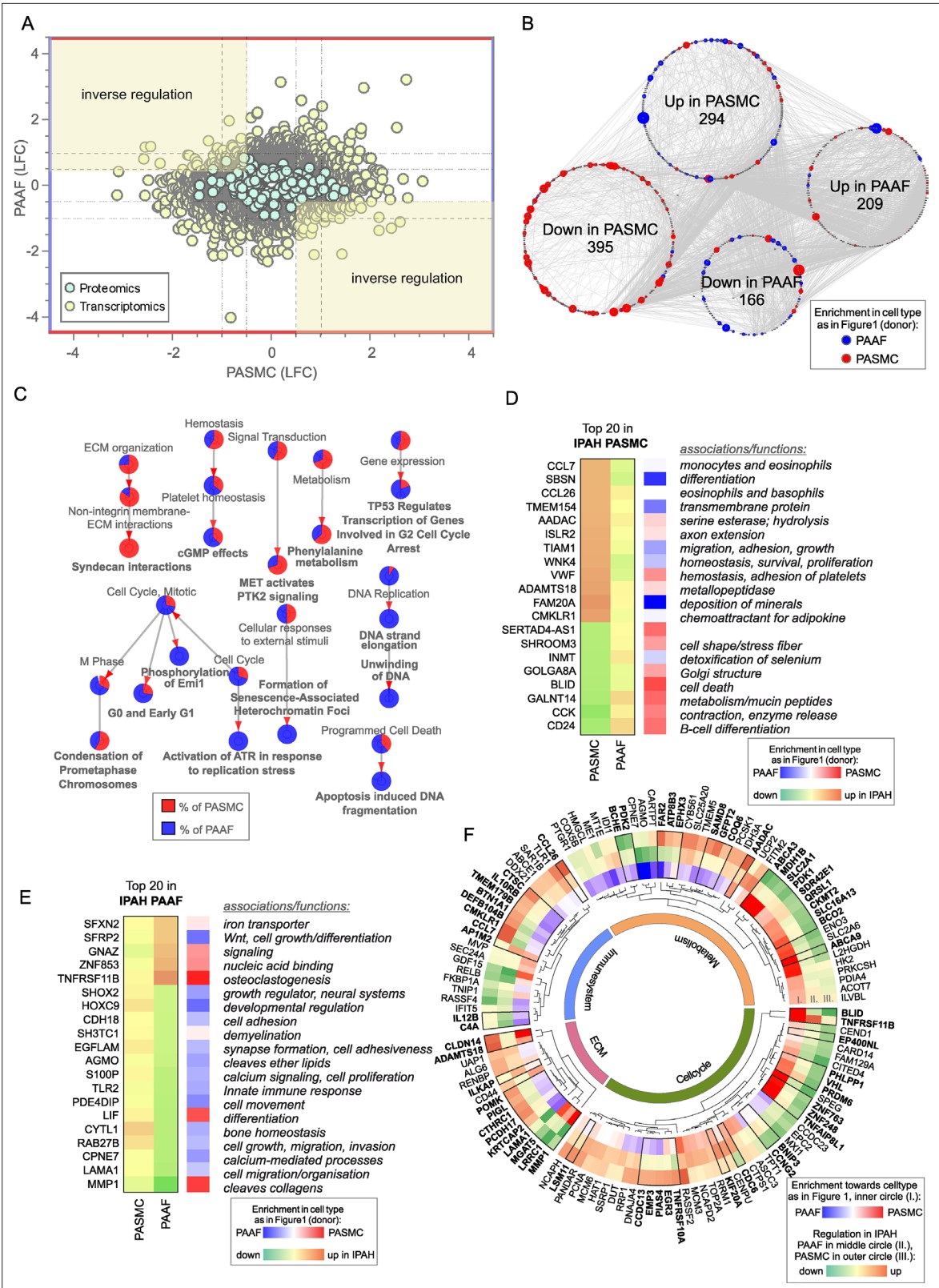

**Figure 5.** Cell-type-specific idiopathic pulmonary arterial hypertension (IPAH)-dependent transcriptomic and proteomic changes. (**A**) Dot plot of $\log_2$ fold changes between donor and IPAH in pulmonary artery smooth muscle cell (PASMC) plotted against changes in pulmonary artery adventitial fibroblast (PAAF), shaded areas highlighting the inversely regulated genes/proteins (criteria employed: $-\log_{10}(p) >1.3$; and LFC ±|1| for transcriptomics data and ±|0.5| for proteomics data). (**B**) A synoptic view of the network analysis performed with all uniquely regulated elements (as highlighted in

*Figure 5 continued on next page*

*Figure 5 continued*

A), depicting their initial enrichment in donor cell type. (**C**) Significant terms resulting from a gene ontology enrichment analysis of uniquely regulated elements represented as nodes connecting to related nodes, reflecting which cell type mostly contributed to the changes observed. Color depicting significantly regulated elements in omics dataset. (**D**) Top 20 most significantly regulated genes in IPAH-PASMC and (**E**) IPAH-PAAF. Color coding reflects the IPAH dependent regulation, and initial enrichment in donor cell type. (**F**) Circular heatmap of IPAH-dependent regulation of metabolic, extracellular matrix, immune system, and cell cycle elements in PASMC and PAAF. The two outer rings give information of direction and intensity of change in IPAH-PASMC (ring panel III) and IPAH-PAAF (ring panel II), while the inner most ring (ring panel I) depicts initial cell-type enrichment at normal condition. Highlights (bold font) are given to significantly differing regulations between PASMC and PAAF under IPAH conditions. The functional association is performed based on data extracted from GeneCards, following a manual curation.

measured attachment and motility properties of cells with respect to these two substrates. Both PASMC and PAAF (*n* = 6) displayed striking phenotypic differences depending on surface coating. Whereas cells on collagen-coated surface had spread-out shape with several protrusions (*Figure 6C*), laminin coating inhibited cell spreading and cells obtained a more symmetric, round-like appearance (*Figure 6C*). PASMC, both donor and IPAH, attached less efficiently to laminin compared to collagen-coated surfaces (*Figure 6D*), but only donor–PAAF displayed comparable behavior (*Figure 6D*). Although not significant, IPAH-PASMC had tendentially higher attachment on laminin-coated surface compared to donor–PASMC (*Figure 6D*). This trend was again observed in an additional assay measuring gap closure on collagen- or laminin-coated surface (*Figure 6E*, *n* = 2–6). At early time points, donor–PASMC showed widening gap area before final gap closure on laminin coating (*Figure 6E*). On collagen coating, potential difference was observed at later time point and showed a reverse trend (*Figure 6E*), probably due to influence of higher proliferative rate of donor–PASMC compared to IPAH-PASMC (*Figure 3E*), rather than acute effects of attachment and migration that have more influence in early time points. On both substrates, PAAF revealed a faster gap closure compared to PASMC (*Figure 6E*), likely in line with their higher proliferative capacity (*Figure 3E*). The expression profile of integrin subunits serving as high-affinity collagen (α2β1) or laminin receptors (α3β1 and α6β1) (*Moiseeva, 2001*) was higher in cultured PASMC compared to PAAF, with minimal change in the disease state (*Figure 6F*). A similar profile was detected in fresh cell, whereby IPAH-PAAF displayed increased expression of alpha2 and alpha6 subunits (*Figure 6G*).

In addition to changes in collagen and laminin class of ECM components, expression profile of proteoglycans showed both cell-type and cell-state distinct pattern with IPAH-PASMC characterized by significant enrichment in this ECM family (*Figure 3B–D*). Among other functions, proteoglycans serve as regulators of the complement system (*Clark et al., 2013*). PAAF display enrichment for complement components such as CFD (*Figure 1E*) and IPAH-PAAF were identified to drive inflammation through classical complement pathway (*Frid et al., 2020*). Two biologically active cleavage products, C3a and C5a, are released in this process and both PASMC and PAAF express corresponding receptors (*Figure 6—figure supplement 1*). We therefore tested a possible feedback loop between the activated complement system and enhanced proteoglycan expression in IPAH. We stimulated donor and IPAH cells with C3a and C5a and measured expression levels of versican (VCAN, PASMC enriched) and decorin (DCN, PAAF enriched) proteoglycans (*Crnkovic et al., 2022*). Both VCAN (*n* = 4) and DCN (*n* = 6) showed a trend or had significantly higher expression level in IPAH cells (*Figure 6H, I*). However, only donor–PAAF responded to C3a and C5a stimulation and displayed IPAH-state-like expression levels of DCN (*Figure 6I*). In summary, these results provide a differential phenotypic readout and link the observed ECM expression changes in a cell-type- and cell-state-specific manner, with PAAF serving generally as sender type and PASMC as responders, excluding the PAAF autocrine complement response.

## IPAH-PAAF modulate phenotypic hallmarks of PASMC

In order to analyze PASMC–PAAF communication in a more comprehensive manner, we took advantage of our published single-cell RNA sequencing dataset from human PA (*Crnkovic et al., 2022*). We extracted PAAF and PASMC clusters and performed intercellular communication network analysis using CellChat. This revealed a skewed PAAF-to-PASMC signaling in IPAH state toward secreted ligands (*Figure 7A*, *Figure 7—figure supplements 1 and 2*). We also performed alternative cell–cell interaction analysis using NicheNet and filtered resulting ligand–receptor pairs for secreted signaling using CellChat database annotation. Merged CellChat and NicheNet analysis results revealed a set

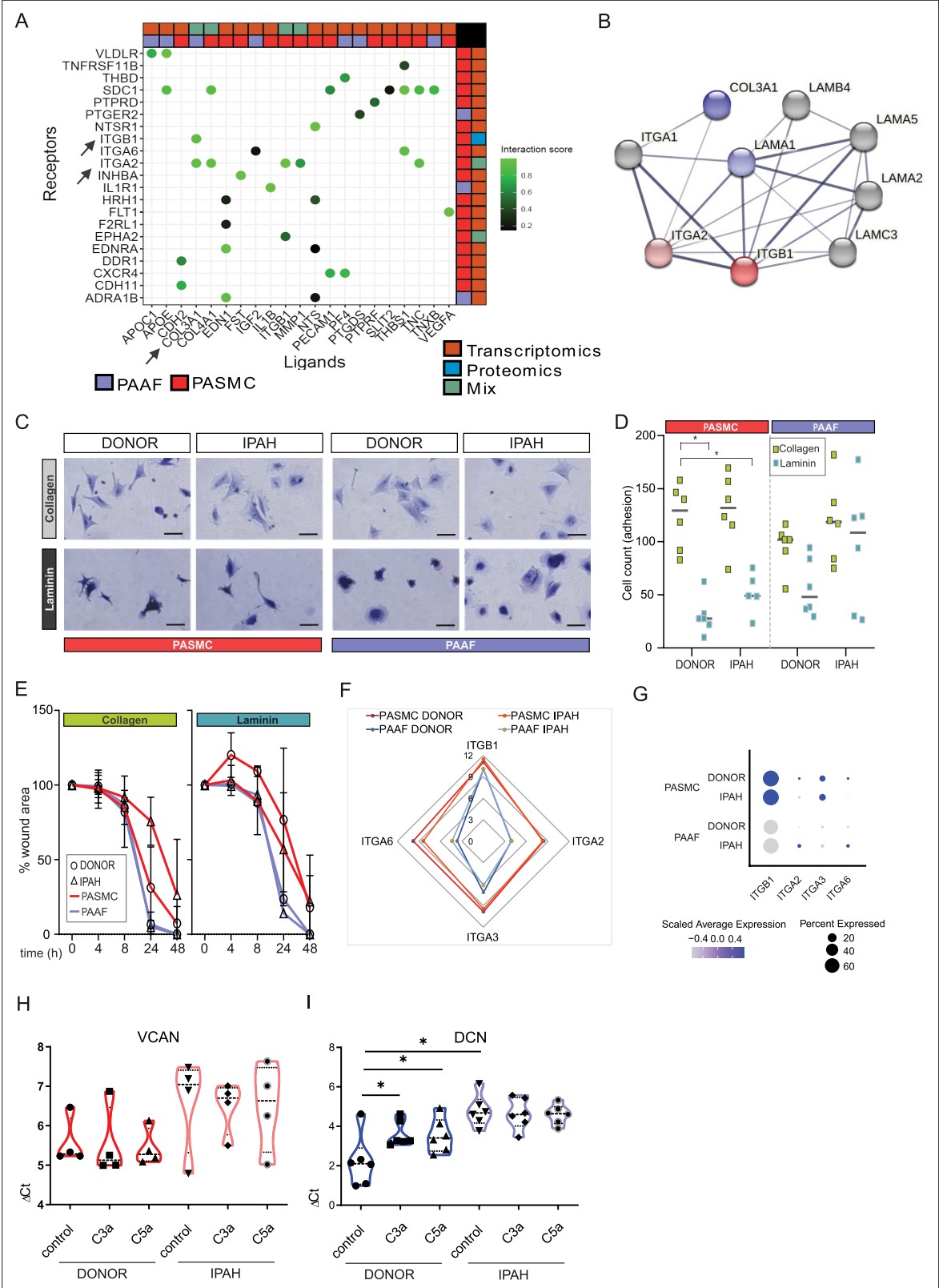

**Figure 6.** Idiopathic pulmonary arterial hypertension (IPAH)-dependent changes mediated through cell-type distinct extracellular matrix response. (**A**) Ligand–receptor interaction analysis for pulmonary artery adventitial fibroblast (PAAF) and pulmonary artery smooth muscle cell (PASMC) based on transcriptomic and proteomic dataset. (**B**) STRING physical interaction network for PASMC expressed receptors (red, ITGA2 and ITGB1) and PAAF expressed ligands (blue, COL3A1 and LAMA1). Edge thickness indicating strength of data support. (**C**) Representative image of crystal violet-stained

*Figure 6 continued on next page*

*Figure 6 continued*

attached cells on collagen-I- or laminin-coated plates. 200 µm scale bar. (**D**) Attachment assay for donor and IPAH-PASMC and PAAF (*n* = 5–6) on collagen-I- and laminin-coated plates (*n* = 6 for each condition). Two-way ANOVA followed by Dunnett's multiple comparisons test (*p < 0.05) for each cell type. (**E**) Gap closure assay on collagen-I (donor/IPAH-PASMC *n* = 3/5, donor/IPAH-PAAF *n* = 5/6)- or laminin-coated (donor/IPAH-PASMC *n* = 2/3, donor/IPAH-PAAF *n* = 6/6) plates. Mean values (presented as % of gap area) over time with standard error mean. (**F**) Mean gene expression values in p1 cells of integrin subunits functioning as collagen and laminin receptors (GSE255669, **Source data 1**). (**G**) Relative expression of integrin subunits in published single-cell transcriptomics dataset of fresh human pulmonary arteries of donors and IPAH patients (GSE210248). (**H**) Versican gene expression in PASMC stimulated 24 hr with active complement components C3a or C5a (100 ng/ml). (**I**) Decorin gene expression in PAAF stimulated 24 hr with active complement components C3a or C5a (100 ng/ml). Mann–Whitney test (*p < 0.05).

The online version of this article includes the following figure supplement(s) for figure 6:

**Figure supplement 1.** Pulmonary artery smooth muscle cell (PASMC) and pulmonary artery adventitial fibroblast (PAAF) expression of anaphylatoxin receptors.

of functionally relevant soluble ligands such as IL1A, POSTN, PDGFA/B, FGF2, BMP2/4, CD40LG perturbed in IPAH state (**Figure 7B**). We therefore asked if cellular crosstalk is able to modulate phenotypic markers using a co-culture model. As reference, we used source-matched donor–PASMC and PAAF (*n* = 3) that were directly co-cultured for 24 hr followed by sorting and quantitative PCR analysis (**Figure 7C**). Expression levels of selected cell-type and cell-state markers were determined as a readout of phenotypic modulation. Simultaneously, same donor cells were co-cultured with IPAH cells from the other cell type, producing a combinatorial readout to assess PASMC-to-PAAF and PAAF-to-PASMC crosstalk (**Figure 7C**). Results revealed that co-culture of donor–PASMC with IPAH-PAAF (*n* = 3) lead to partial loss of healthy state PASMC markers (CNN1 and RHOA) without affecting the expression of disease-state markers (VCAN and BGN) (**Figure 7D**). Expression of neither healthy nor disease-state PAAF markers was affected in donor–PAAF co-cultured with IPAH-PASMC (*n* = 3) (**Figure 7D**). To get a better understanding on donor–PASMC state shift induced by co-culture with IPAH-PAAF, we performed bulk RNA-Seq on collected samples. As the effect of 24 hr co-culture was mild compared to sample-driven variability, we performed model correction to account for paired PASMC samples in our experimental setup. The resulting GSEA on significantly regulated genes (**Source data 4**) showed that genes upregulated in PASMC by IPAH-PAAF co-culture mediate inflammatory processes and signaling, apoptosis, and proliferation (**Figure 7E**).

We next went on to identify possible PAAF ligands that could drive the observed phenotypic change in PASMC. As our proteomics approach sampled only proteins from the cell lysate, we additionally performed protein array-based screening of 137 unique soluble factors secreted from donor- compared to IPAH-PAAF (*n* = 4) (**Figure 7F**, **Figure 7—figure supplement 3**). We selected for factors whose expression levels were changed 10% or more in at least two samples and performed clustering of tested samples and factors providing a list of 44 potential regulators (**Figure 7G**). As some of those factors were already known and investigated in the context of IPAH, we focused on finding potentially novel PAAF-derived regulators of SMC phenotype. In order to narrow down the search, we used hierarchical clustering to group co-regulated ligands into smaller number of groups. Similarity matrix revealed one major and six minor clusters of factors showing correlated changes among four comparison pairs (**Figure 7—figure supplement 3**). One cluster was characterized by co-regulated chitinase 3-like 1, insulin-like growth factor-binding protein 3 (IGFBP-3), and angiogenin expression pattern; other clusters with Extracellular Matrix Metalloproteinase Inducer (EMMPRIN), Epithelial-Derived Neutrophil-Activating Protein 78 (ENA-78), and Vascular Endothelial Growth Factor A (VEGF) secretion, respectively (**Figure 7—figure supplement 3**). Three additional clusters involved upregulated secretion of epidermal growth factor, HGF, and chemerin with correlated PTX3 downregulation (**Figure 7—figure supplement 3**). Based on further literature search, we excluded ligands with reported link to PAH and selected chitinase 3-like 1, IGFBP-3, angiogenin, HGF, and PTX3 for further validation. Stimulation of donor and IPAH-PASMC (*n* = 6) with HGF and PTX3 displayed the most consistent effects on modulation of PASMC cell-state markers (**Figure 7H**), while other tested ligands mostly lacked the effect on donor–PASMC and were not investigated further (**Figure 7—figure supplement 4**). Intriguingly, HGF and PTX3 seem to have opposite effect on vascular smooth muscle cells, with PTX3 inhibiting neointima formation (**Camozzi et al., 2005**) and HGF promoting pulmonary vascular remodeling (**Park et al., 2022**). Indeed, stimulation of PASMC with PTX3 preserved the

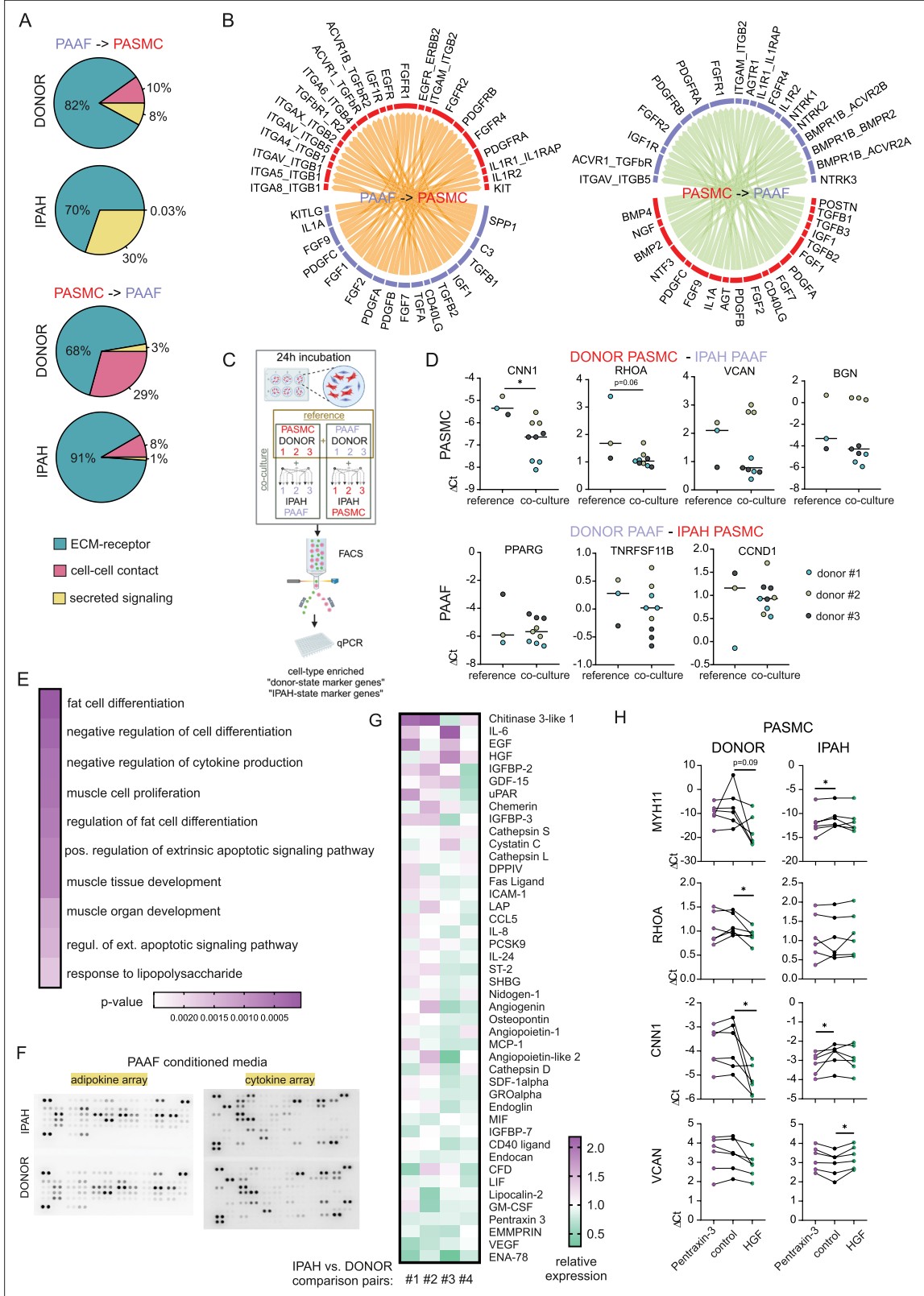

**Figure 7.** Skewed expression profile of pulmonary artery smooth muscle cell (PASMC)-state markers mediated by idiopathic pulmonary arterial hypertension (IPAH)-dependent changes in pulmonary artery adventitial fibroblast (PAAF) ligands. (**A**) Ligand–receptor interaction analysis for PAAF and PASMC based on single-cell RNA sequencing dataset. (**B**) Circos plot showing consensus soluble ligand and cognate receptor pairs identified by CellChat and NicheNet analysis of PAAF or PASMC as sender (ligand-expressing) cell type. (**C**) Schematic representation of PASMC–PAAF co-culture

experiment to determine the influence of IPAH cells on phenotypic marker expression in donor cells. Created wiht BioRender.com. (**D**) Gene expression of health (calponin: CNN1, Ras Homolog Family Member A: RHOA, peroxisome proliferator-activated receptor gamma: PPARG, osteoprotegerin: TNRFSF11B) and disease (versican: VCAN, biglycan: BGN, cyclin D1: CCDN1) state phenotypic markers in donor–PASMC (upper row) and donor–PAAF (lower row) following their co-culture with IPAH cells. (**E**) Top 10 gene ontology biological processes enriched in the set of significantly upregulated genes in donor–PASMC co-cultured with IPAH-PAAF compared to reference (source-matched donor–PAAF). (**F**) Representative protein array scans determining the content of soluble ligands secreted by donor or IPAH-PAAF over 24 hr in cell culture medium. (**G**) Heatmap of fold change expression of soluble ligands secreted by IPAH-PAAF compared to donor–PAAF (*n* = 4 for each condition). (**H**) Gene expression changes of cell-state markers in donor and IPAH-PASMC (*n* = 6) treated 24 hr with pentraxin-3 (5 μg/ml) or hepatocyte growth factor (HGF, 25 ng/ml). Smooth muscle myosin heavy chain: MHY11. Friedman test followed by Dunn's multiple comparisons test (*p < 0.05).

The online version of this article includes the following figure supplement(s) for figure 7:

**Figure supplement 1.** Differential cell–cell communication analysis with pulmonary artery adventitial fibroblast (PAAF) as sender cell type.

**Figure supplement 2.** Differential cell–cell communication analysis with pulmonary artery smooth muscle cell (PASMC) as sender cell type.

**Figure supplement 3.** Soluble factor measurements from pulmonary artery adventitial fibroblasts (PAAF) cell culture supernatants.

**Figure supplement 4.** Gene expression profile of pulmonary artery smooth muscle cell (PASMC)-state markers upon stimulation with additional pulmonary artery adventitial fibroblast (PAAF) ligands.

expression of contractile markers in donor cells, but not in IPAH, while HGF downregulated contractile state markers in donor–IPAH and increased disease-state marker VCAN in IPAH-PASMC (*Figure 7H*).

## Discussion

Effective treatment of pulmonary arterial remodeling has been elusive over the years (*Stacher et al., 2012*). IPAH is characterized by recalcitrant morphological changes and persistent alterations in cellular function. Studies have indicated that PASMC and PAAF retain these disease-associated changes in in vitro conditions (*Stenmark et al., 2018*; *Viswanathan et al., 2023*; *Zhang et al., 2017*). Our current results using transcriptomic and proteomic profiling in combination with phenotypic screening demonstrate that PASMC and PAAF acquire distinct IPAH-associated cell states, while retaining their original cell-type designation. This shift from healthy to diseased state is a cell-type-specific process and results with functionally meaningful differences between cell types. Furthermore, we identify that maintenance of PASMC cell state is dependent on external cues, such as ECM components and soluble ligands, provided in part by neighboring PAAF.

The divergent fates and distinct disease states that cells acquire, can be traced to initial cell lineage differences that are formed and imprinted in a tissue- and anatomical location-specific manner (*Muhl et al., 2022*; *Muhl et al., 2020*). These anatomical-location-based distinctions provide fibroblast and smooth muscle cell lineages with phenotypic potential beyond canonically assigned roles.

Expression of three selected markers in PAAF, scavenger receptor class A member 5 (SCARA5), complement factor D (CFD) and alcohol dehydrogenase 1C (ADH1C), provide the best example of PAAF pleiotropic roles beyond mere ECM production and structural support. PAAF, being at the outer barrier of vessels, might act as innate immunity sentinels with adventitia being the main compartment of inflammatory reaction and homing site for inflammatory cells (*Dahlgren et al., 2019*). CFD activates the alternative pathway of the complement triggering chemotaxis and inflammation (*White et al., 1992*). SCARA5, a ferritin scavenger receptor that mediates cellular iron uptake, limits the extra-cellular iron concentration (*Li et al., 2009*). Going beyond putative immunomodulatory functions, we additionally reveal PAAF involvement in lipid metabolism at expression profile and phenotypic level. This functionality perhaps reflects broader ontogenetic potential of PDGFRA+ cells as adipocyte precursors (*Angueira et al., 2021*): PAAF express ADH1C, enzyme involved in metabolism of lipid peroxidation products and hydroxysteroids (*Höög and Ostberg, 2011*), PPARG, a key transcriptional regulator of lipid metabolism, and APOE, a lipid handling protein (*Source data 1* and *Source data 2*). All three homeostatic functions, complement regulation, iron handling, and lipid metabolism, seem to be co-opted and distorted in PAH pathomechanism (*Frid et al., 2020*; *Ruiter et al., 2011*; *Zhang et al., 2017*).

Simultaneously, acquisition of disease-cell state by the PASMC follows a different course. First, we observed that already healthy PASMC display intrinsically higher heterogeneity compared to PAAF. Enrichment in developmental, morphogenesis, migration, and signal transduction biological

processes coupled with higher metabolic rate, both oxidative and glycolytic, provides PASMC with functional flexibility and adaptability in response to various injuries. Expression of regulator of G protein signaling 5 (RGS5) provides an example of the plasticity potential imprinted in normal PASMC. RGS5, for instance, as a gatekeeper of G protein signaling and signal transduction modulator, influences and fine-tunes the PASMC response to external cue (*Wang et al., 2008*). Under normal condition, expression of contractile machinery coupled with higher metabolic rate, enables PASMC to fulfill their homeostatic role in providing vasoactive function and structural support for endothelial lined lumen. However, one of the defining vascular SMC characteristic and adaptation to vascular insult, including that of IPAH-PASMC cell state, is downregulated expression of the cytoskeletal contractile elements. This seems to be an intrinsic, 'programmed' response of SMC to remodeling, rather than vasodilator therapy-induced selection pressure. Similar phenotypic change is observed in SMC from systemic circulation (*Gomez and Owens, 2012*) and animal model of pulmonary hypertension without exposure to PAH medication (*Crnkovic et al., 2023*). We suspect that isolated decrease in contractile machinery would dampen both the contraction and relaxation properties of the PASMC, affecting not only its response to dilating agents, but also to vasoconstrictors. Clinical consequences and responsiveness to dilating agents are more difficult to predict, since the vasoactive response would additionally depend on mechanical properties of the pulmonary artery defined by cellular and ECM composition (*Thenappan et al., 2018*).

Distinctive cell-type characteristics of PASMC and PAAF come into play during disease process of pulmonary vascular remodeling, testified by low overlap in commonly perturbed pathways and associated genes/proteins between PASMC and PAAF in IPAH. PAAF and PASMC acquire distinct cell states upon remodeling with IPAH-PAAF showing enrichment in DNA replication and repair processes and accompanying hyperproliferative phenotype. IPAH-PASMC in addition to partial loss of contractile markers alter their metabolic profile to support the increased biosynthetic demand. In line with this notion, we have observed significant downregulation of enolase 3, malate dehydrogenase 1B, pyruvate dehydrogenase kinases, and phosphofructokinase, and upregulation of glutamine-fructose-6-phosphate transaminase 2 in IPAH-PASMC (*Source data 1* and *Source data 2*). This profile indicated potential glucose shunting into hexosamine pathway (*Barnes et al., 2015*; *Buse, 2006*). Hexosamine biosynthetic pathway is a side branch of glycolysis utilized for generation of amino-sugars and subsequent protein and lipid glycosylation, including glycosaminoglycans and PG synthesis (*Buse, 2006*) The mostly enriched ECM class next to classical fibrillar collagens, were PG, in line with previously published reports (*Barnes et al., 2015*; *Chang et al., 2016*; *Crnkovic et al., 2022*; *Jandl et al., 2020*; *Mutgan et al., 2024*). Due to the complex structure of PG, they can withstand high pressure, modulate cellular phenotype and thus be a necessary part of the adaptation or maladaptation process (*Wight and Merrilees, 2004*).

Despite broad changes in expression profiles in both cell types upon vascular remodeling and strikingly low overlap in perturbed pathways between PASMC and PAAF, they nevertheless share one common feature. Expression pattern of key cell-type-enriched elements is preserved in the diseased state. This finding is further supported by the global analysis of transcriptomic and proteomic profiles revealing that cell type as a variable has a stronger influence on sample distinction then disease condition. In other words, majority of IPAH-PASMC are still more alike to donor–PASMC then to any PAAF.

Interestingly, both cell types share a common disturbance in mitochondria characterized by lower IPAH mitochondrial content and hyperpolarized mitochondrial membrane potential. Nevertheless, ultimate phenotypic consequences of such a change still seems to be divergent for each cell type, as implied by distinct stimulated ROS response. Whether mitochondrial disturbance is one the initial cause of the disease or its consequence, as a common point it offers a potential action point to revert cell states from different lineages toward a normal, homeostatic condition, in line with reports on restoring SOD2 expression as a viable therapeutic target (*Paulin et al., 2014*; *Plecitá Hlavatá et al., 2023*).

Going beyond cataloguing molecular and functional changes, we address the underlying communication mechanisms responsible for maintenance and transition between identified cell states. We first dissect the role of ECM composition and reveal how a relative change in the laminin-to-collagen ratio, driven by a shift in PAAF cell state, similarly alters the spreading, attachment, migration, and expansion of both normal and diseased PASMCs. The comparable responses of donor and IPAH-PASMC likely result from their shared integrin receptor expression profiles. Meanwhile, ECM class

switching engages different high-affinity integrin receptors, which activate alternative downstream signaling pathways (*Nguyen et al., 2000*) and lead to differential responses to collagen and laminin matrices. We thus propose a model in which laminins and collagens act as PAAF-secreted ligands, regulating PASMC behavior through their ECM-sensing integrin receptors.

We also uncover an autocrine loop centered on PAAF and showing how activated complement pathway can drive expression of PAAF-enriched ECM components. These processes could induce a more invasive cellular phenotype and potentially promote neointima formation. One could trace this PASMC susceptibility to ECM composition change to their enriched expression in integrin subunits serving as collagen and laminin receptors, making them uniquely primed to sense and respond to external cues. However, the same mechanism renders cells susceptible to phenotypic change induced simply by extended in vitro culturing, testified by broad expression profile differences between fresh and cultured cells. This common caveat in cell biology research and represents a technical and practical tradeoff that requires cross validation of key findings. Using a combination of archived lung tissue and available single-cell RNA sequencing dataset of human pulmonary arteries, we show that some of the key defining phenotypic features of diseased cells, such as altered proliferation rate and ECM production, are preserved and gradually lost upon prolonged culturing. Interestingly, this culturing effect was more pronounced in PAAF particularly with respect to inverse direction of enrichment for inflammatory/cytokine signaling process. The removal of PAAF from their native immunomodulatory environment and the presence of hydrocortisone in PAAF culture media could provide a potential explanation for this effect.

We extend our investigations more broadly and uncover a general shift in IPAH-PAAF secretome being sufficient to induce significant phenotypic changes in normal PASMC. Among several deregulated ligands, we identify downregulated expression of PTX3, a protective factor, and upregulation of HGF, pathogenic factor, as PAAF-sourced regulators of PASMC cell state. PTX3 counteracts pro-proliferative action of fibroblast growth factor and its overexpression attenuates neointima formation in systemic arteries (*Camozzi et al., 2005*). Conversely, upregulation of HGF/cMet pathway in endothelial cells upon Sox17 loss was associated with PAH promotion (*Park et al., 2022*). We would therefore argue that potential exploratory clinical trial for pulmonary vascular disease could entail a combination therapy consisting of recombinant PTX3 and HGF blocking antibody or c-Met inhibitor. However, the identified factors likely represent just a fraction of ligands expressed by other cell types present in the pulmonary artery compartment. The influence of endothelial and inflammatory cells on PASMC and PAAF phenotype requires further investigation.

A low case number and end-stage disease samples used for omics characterization represents a study limitation that has to be taken into account before assuming similar findings would be evident in the entire PAH patient population over the course of the disease development and progression.

Nevertheless, our results provide detailed understanding of functional changes in relation to underlying molecular expression differences upon pulmonary vascular remodeling. Furthermore, we identify novel targets for cell-type-specific correction of diseased cell states and potential reverse remodeling strategies.

## Acknowledgements

We appreciate excellent technical assistance from Julia Kohlbacher, Hans Peter Ziegler, and Elisabeth Blanz. For open access purposes, the author has applied a CC BY public copyright license to any author accepted manuscript version arising from this submission.

## Additional information

### Funding

| Funder | Grant reference number | Author |
| --- | --- | --- |
| Cardio-Pulmonary Institute | EXC 2026 390649896 | Grazyna Kwapiszewska |
| Austrian Science Fund | 10.55776/i4651 | Slaven Crnkovic |

| Funder | Grant reference number | Author |
|---|---|---|

The funders had no role in study design, data collection and interpretation, or the decision to submit the work for publication.

## Author contributions

Slaven Crnkovic, Conceptualization, Data curation, Formal analysis, Investigation, Visualization, Methodology, Writing – original draft, Writing – review and editing; Helene Thekkekara Puthenparampil, Conceptualization, Formal analysis, Investigation, Visualization, Writing – original draft; Shirin Mulch, Nemanja Radic, Data curation, Visualization, Methodology; Valentina Biasin, Bence Miklos Nagy, Data curation, Formal analysis, Investigation; Jochen Wilhelm, Conceptualization, Data curation, Formal analysis, Visualization; Marek Bartkuhn, Data curation, Formal analysis, Visualization, Methodology; Ehsan Bonyadi Rad, Formal analysis, Investigation; Alicja Wawrzen, Formal analysis, Investigation, Methodology, Writing – review and editing; Ingrid Matzer, Formal analysis, Investigation, Methodology; Ankita Mitra, Data curation, Formal analysis, Methodology; Ryan D Leib, Resources, Formal analysis, Investigation; Anita Sahu-Osen, Investigation, Visualization, Methodology, Writing – original draft; Francesco Valzano, Natalie Bordag, Data curation, Formal analysis, Visualization; Matthias Evermann, Resources, Data curation, Methodology; Konrad Hoetzenecker, Resources, Data curation, Methodology, Project administration; Andrea Olschewski, Writing – review and editing; Senka Ljubojevic-Holzer, Malgorzata Wygrecka, Investigation, Methodology; Kurt Stenmark, Leigh M Marsh, Conceptualization, Methodology, Writing – review and editing; Vinicio de Jesus Perez, Conceptualization, Data curation, Supervision, Funding acquisition, Investigation, Visualization, Project administration, Writing – review and editing; Grazyna Kwapiszewska, Conceptualization, Resources, Data curation, Supervision, Funding acquisition, Investigation, Writing – original draft, Project administration, Writing – review and editing

## Author ORCIDs

Slaven Crnkovic (iD) https://orcid.org/0000-0002-0820-3318
Jochen Wilhelm (iD) https://orcid.org/0000-0001-5544-9647
Malgorzata Wygrecka (iD) https://orcid.org/0000-0002-3656-2932
Leigh M Marsh (iD) https://orcid.org/0000-0002-1754-9249
Vinicio de Jesus Perez (iD) https://orcid.org/0000-0001-5532-8247
Grazyna Kwapiszewska (iD) https://orcid.org/0000-0003-0518-9079

## Ethics

Human lung samples from IPAH patients and downsized non-transplanted donor lungs, serving as healthy control, were obtained from Division of Thoracic Surgery, Medical University of Vienna, Austria. The protocol and tissue usage was approved by the local ethics committee (976/2010; 1417/2022) and patient consent was obtained before lung transplantation.

Reviewer #1 (Public review): https://doi.org/10.7554/eLife.98558.3.sa1
Reviewer #2 (Public review): https://doi.org/10.7554/eLife.98558.3.sa2
Author response https://doi.org/10.7554/eLife.98558.3.sa3

# Additional files

## Supplementary files

Supplementary file 1. Patient characteristics. Age and sex of healthy controls (donors) and patients with pulmonary vascular disease (IPAH) with corresponding clinical data (mean pulmonary arterial pressure, mPAP, cardiac output) and PAH therapy.

Supplementary file 2. List of antibodies and detection reagents.

Supplementary file 3. List of used primers.

Source data 1. Differential gene expression analysis table in source-matched very early passage PASMC and PAAF from donors and IPAH patients.

Source data 2. Differential expression analysis of mass spectrometric measurement proteins in source-matched donor and IPAH PASMC and PAAF.

Source data 3. List of significantly enriched gene ontology biological process terms in donor and IPAH PASMC and PAAF cells in fresh, very early and later passage cells.

Source data 4. Differential gene expression analysis of donor-PASMC co-cultured with donor-PAAF compared with donor-PASMC co-cultured with IPAH-PAAF.

MDAR checklist

## Data availability

Genome-wide expression profiling and RNA-Seq data are deposited to the National Center of Biotechnology Information Gene Expression Omnibus database (accession numbers GSE255669, GSE262069). Human tissue samples can be provided pending ethical approval and completed material transfer agreement.

The following datasets were generated:

| Author(s) | Year | Dataset title | Dataset URL | Database and Identifier |
|---|---|---|---|---|
| Kwapiszewska G, Wilhelm J | 2024 | IPAH signatures in hPASMC and hAdvFB | https://www.ncbi.nlm.nih.gov/geo/query/acc.cgi?acc=GSE255669 | NCBI Gene Expression Omnibus, GSE255669 |
| Kwapiszewska G, Crnkovic S, Bartkuhn M | 2024 | Adventitial fibroblasts direct smooth muscle cell-state transition in pulmonary vascular disease | https://www.ncbi.nlm.nih.gov/geo/query/acc.cgi?acc=GSE262069 | NCBI Gene Expression Omnibus, GSE262069 |

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
