## [Editor Report · eLife Assessment]

This **fundamental** research conducted a molecular comparison between smooth muscle cells and adjacent fibroblast cells within lung blood vessels affected by pulmonary arterial hypertension. The study identified distinct disease-related states in each cell type and provided deeper insights into their interactions and communication. While certain conclusions should be interpreted with caution due to inherent methodological limitations, the study's findings remain **convincing** and robust. This is supported by the use of advanced and complementary techniques, as well as the rare isolation of diseased lung blood vessel cells from the same donor, enabling direct comparison.

---

## [Referee Report · Reviewer #1 (Public review)]

Summary:

The authors isolated and cultured pulmonary artery smooth muscle cells (PASMC) and pulmonary artery adventitial fibroblasts (PAAF) of the lung samples derived from the patients with idiopathic pulmonary arterial hypertension (PAH) and the healthy volunteers. They performed RNA-seq and proteomics analyses to detail the cellular communication between PASMC and PAAF, which are the main target cells of pulmonary vascular remodeling during the pathogenesis of PAH. The authors revealed that PASMC and PAAF retained their original cellular identity and acquired different states associated with the pathogenesis of PAH, respectively.

Strengths:

Although previous studies have shown that PASMC and PAAF cells each have an important role in the pathogenesis of PAH, there have been scarce reports focusing on the interactions between PASMC and PAAF. These findings may provide valuable information for elucidating the pathogenesis of pulmonary arterial hypertension.

Comments on revisions:

The authors adequately responded to my concerns and revised their manuscript to elaborate on the new data from new experiments and address my queries. Although some of the issues I initially raised could not be fully resolved, the revised manuscript has been significantly improved. This manuscript provides essential insights into the communications across the PASMCs and PAAFs in PAH. This would greatly interest various researchers in both basic and clinical fields.

---

## [Referee Report · Reviewer #2 (Public review)]

Summary:

Utilizing a combination of transcriptomic and proteomic profiling as well as cellular phenotyping from source-matched PASMC and PAAFs in IPAH, this

study sought to explore a molecular comparison of these cells in order to track distinct cell fate trajectories and acquisition of their IPAH-associated cellular states. The authors also aimed to identify cell-cell communication axes in order to infer mechanisms by which these two cells interact and depend upon external cues. This study will be of interest to the scientific and clinical communities of those interested in pulmonary vascular biology and disease. It also will appeal to those interested in lung and vascular development as well as multi-omic analytic procedures.

Strengths:

(1) This is one of the first studies using orthogonal sequencing and phenotyping for characterization of source-matched neighoring mesenchymal PASMC and PAAF cells in healthy and diseased IPAH patients. This is a major strength which allows for direct comparison of neighboring cell types and the ability to address an unanswered question regarding the nature of these mesenchymal "mural" cells at a precise molecular level.

(2) Unlike a number of multi-omic sequencing papers that read more as an atlas of findings without structure, the inherent comparative organization of the study and presentation of the data were valuable in aiding the reader in understanding how to discern the distinct IPAH-associated cell states. As a result, the reader not only gleans greater insight into these two interacting cell types in disease but also now can leverage these datasets more easily for future research questions in this space.

(3) There are interesting and surprising findings in the cellular characterizations, including the low proliferative state of IPAH-PASMCs as compared to the hyperproliferative state in IPAH-PAAFs. Furthermore, the cell-cell communication axes involving ECM components and soluble ligands provided by PAAFs that direct cell state dynamics of PASMCs offer some of the first and foundational descriptions of what are likely complex cellular interactions that await discovery.

(4) Technical rigor is quite high in the -omics methodology and in vitro phenotyping tools used.

Weaknesses:

There are some weaknesses in the methodology that should temper the conclusions:

(1) The number of donors sampled for PAAF/PASMCs was relatively small for both healthy controls and IPAH patients. Thus, while the level of detail of -omics profiling was quite deep, the generalizability of their findings to all IPAH patients or Group 1 PAH patients is limited. In the revised manuscript, the authors addressed this concern with important text changes and additional data.

(2) While the study utilized early passage cells, these cells nonetheless were still cultured outside the in vivo milieu prior to analysis. Thus, while there is an assumption that these cells do not change fundamental behavior outside the body, that is not entirely proven for all transcriptional and proteomic signatures. As such, the major alterations that are noted would be more compelling if validated from tissue or cells derived directly from in vivo sources. Without such validation, the major limitation of the impact and conclusions of the paper is that the full extent of the relevance of these findings to human disease is not known. The authors addressed this concern appropriately with significant text changes to clarify these limitations for the reader.

(3) While the presentation of most of the manuscript was quite clear and convincing, the terminology and conclusions regarding "cell fate trajectories" throughout the manuscript did not seem to be fully justified. That is, all of the analyses were derived from cells originating from end-stage IPAH, and otherwise, the authors were not lineage tracing across disease initiation or development (which would be impossible currently in humans). So, while the description of distinct "IPAH-associated states" makes sense, any true cell fate trajectory was not clearly defined. The revised manuscript has removed this terminology and replaced it with more precise language.

Comments on revisions:

The authors were quite responsive to all of my concerns, offering both important revisions to the presentation of the work as well as new data. While some of the limitations were not fully resolved (and the authors provide appropriate justification for this), the revised manuscript is much improved. It will be of great interest to both the scientific and clinical communities.

---

## [Author Response]

The following is the authors’ response to the original reviews.

**eLife assessment**
This important study explored a molecular comparison of smooth muscle and neighboring fibroblast cells found in lung blood vessels afflicted by a disease called pulmonary arterial hypertension. In doing so, the authors described distinct disease-associated states of each of these cell types with further insights into the cellular communication and crosstalk between them. The strength of evidence was convincing through the use of complementary and sophisticated tools, accompanied by rare isolation of human diseased lung blood vessel cells that were source-matched to the same donor for direct comparison.

We thank the editors and reviewers in their highly positive and encouraging assessment of our manuscript detailing the cell state changes of arterial smooth muscle cells and fibroblasts in the pulmonary bed. We addressed reviewers’ major comments in the revised manuscript by providing validation of key in vitro findings, such as preserved marker localization and increased GAG deposition in IPAH pulmonary arteries. We additionally provide comparison of transcriptomic profiles spanning fresh, very early and late passage cells. Finally, we present expanded experimental data in support of cellular crosstalk, including testing of additional PAAF ligands on donor PASMC and influence of PTX3/HGF on IPAH PASMC.

**Public Reviews:**

**Reviewer #1 (Public Review):**
Summary:The authors isolated and cultured pulmonary artery smooth muscle cells (PASMC) and pulmonary artery adventitial fibroblasts (PAAF) of the lung samples derived from the patients with idiopathic pulmonary arterial hypertension (PAH) and the healthy volunteers. They performed RNA-seq and proteomics analyses to detail the cellular communication between PASMC and PAAF, which are the main target cells of pulmonary vascular remodeling during the pathogenesis of PAH. The authors revealed that PASMC and PAAF retained their original cellular identity and acquired different states associated with the pathogenesis of PAH, respectively.Strengths:Although previous studies have shown that PASMC and PAAF cells each have an important role in the pathogenesis of PAH, there have been scarce reports focusing on the interactions between PASMC and PAAF. These findings may provide valuable information for elucidating the pathogenesis of pulmonary arterial hypertension.

We appreciate the reviewer’s positive view of our study.

Weaknesses:The results of proteome analysis using primary culture cells in this paper seem a bit insufficient to draw conclusions. In particular, the authors described "We elucidated the involvement of cellular crosstalk in regulating cell state dynamics and identified pentraxin-3 and hepatocyte growth factor as modulators of PASMC phenotypic transition orchestrated by PAAF." However, the presented data are considered limited and insufficient.

We thank the reviewer for drawing our attention to this point and have accordingly modified the conclusion section to read: “We investigated the involvement of cellular crosstalk….” Moreover, we provide further experimental evidence demonstrating the effect of both PTX3 and HGF on cell state marker expression in IPAH-PASMC cells (Figure 7H). In addition, we clarify the selection strategy applied to investigate particular PAAF-secreted ligands and test three additional ligands on donor PASMC (Figure S8), supporting the original focus on PTX3 and HGF.

**Reviewer #2 (Public Review):**
Summary:Utilizing a combination of transcriptomic and proteomic profiling as well as cellular phenotyping from source-matched PASMC and PAAFs in IPAH, this study sought to explore a molecular comparison of these cells in order to track distinct cell fate trajectories and acquisition of their IPAH-associated cellular states. The authors also aimed to identify cell-cell communication axes in order to infer mechanisms by which these two cells interact and depend upon external cues. This study will be of interest to the scientific and clinical communities of those interested in pulmonary vascular biology and disease. It also will appeal to those interested in lung and vascular development as well as multi-omic analytic procedures.

We thank the reviewer for overall highly positive assessment of our study.

Strengths:(1) This is one of the first studies using orthogonal sequencing and phenotyping for the characterization of source-matched neighboring mesenchymal PASMC and PAAF cells in healthy and diseased IPAH patients. This is a major strength that allows for direct comparison of neighboring cell types and the ability to address an unanswered question regarding the nature of these mesenchymal "mural" cells at a precise molecular level.

We value the reviewer’s kind and objective summary of our study.

(2) Unlike a number of multi-omic sequencing papers that read more as an atlas of findings without structure, the inherent comparative organization of the study and presentation of the data were valuable in aiding the reader in understanding how to discern the distinct IPAH-associated cell states. As a result, the reader not only gleans greater insight into these two interacting cell types in disease but also now can leverage these datasets more easily for future research questions in this space.

We thank the reviewer for this highly positive comment.

(3) There are interesting and surprising findings in the cellular characterizations, including the low proliferative state of IPAH-PASMCs as compared to the hyperproliferative state in IPAH-PAAFs. Furthermore, the cell-cell communication axes involving ECM components and soluble ligands provided by PAAFs that direct cell state dynamics of PASMCs offer some of the first and foundational descriptions of what are likely complex cellular interactions that await discovery.

We agree with the reviewer’s assessment that some of the novel data in our study helps to formulate testable hypothesis that can be followed through with more focused follow-up research.

(4) Technical rigor is quite high in the -omics methodology and in vitro phenotyping tools used.

We are grateful for reviewer’s assessment of our work and positive recognition.

Weaknesses:There are some weaknesses in the methodology that should temper the conclusions:(1) The number of donors sampled for PAAF/PASMCs was small for both healthy controls and IPAH patients. Thus, while the level of detail of -omics profiling was quite deep, the generalizability of their findings to all IPAH patients or Group 1 PAH patients is limited.

We appreciate the reviewers concerns regarding the generalizability of the findings and have acknowledged this as the study limitation in the discussion: “A low case number and end-stage disease samples used for omics characterization represents a study limitation that has to be taken into account before assuming similar findings would be evident in the entire PAH patient population over the course of the disease development and progression”. We have addressed this issue by performing validation of key in vitro findings using fresh cells or assessment of FFPE lung material from additional independent samples in the revised manuscript (Figures 2D, 3D, 3H, 4H). For transparency, we provide biological sample number in the result section of the modified manuscript.

(2) While the study utilized early passage cells, these cells nonetheless were still cultured outside the in vivo milieu prior to analysis. Thus, while there is an assumption that these cells do not change fundamental behavior outside the body, that is not entirely proven for all transcriptional and proteomic signatures. As such, the major alterations that are noted would be more compelling if validated from tissue or cells derived directly from in vivo sources. Without such validation, the major limitation of the impact and conclusions of the paper is that the full extent of the relevance of these findings to human disease is not known.

We thank the reviewer for this constructive and excellent suggestion. The comparison of fresh and cultured cells revealed a strong and early divergence of differentially regulated pathways for PAAF, while a more gradual transition for PASMC. The results of this analysis are included in the new Figures 2D, 3D, 3H, and 4H. Implications are discussed in the revised manuscript: “However, the same mechanism renders cells susceptible to phenotypic change induced simply by extended vitro culturing, testified by broad expression profile differences between fresh and cultured cells. This common caveat in cell biology research and represents a technical and practical tradeoff that requires cross validation of key findings. Using a combination of archived lung tissue and available single cell RNA sequencing dataset of human pulmonary arteries, we show that some of the key defining phenotypic features of diseased cells, such as altered proliferation rate and ECM production, are preserved and gradually lost upon prolonged culturing”.

(3) While the presentation of most of the manuscript was quite clear and convincing, the terminology and conclusions regarding "cell fate trajectories" throughout the manuscript did not seem to be fully justified. That is, all of the analyses were derived from cells originating from end-stage IPAH, and otherwise, the authors were not lineage tracing across disease initiation or development (which would be impossible currently in humans). So, while the description of distinct "IPAH-associated states" makes sense, any true cell fate trajectory was not clearly defined.

In accordance to reviewer’s comment, we have decided to modify the wording to exclude the “cell fate trajectory” phrase and replace it with “acquisition of disease cell state”.

**Recommendations for the authors:**

**Reviewer #1 (Recommendations For The Authors):**
Major comments:(1) In Figure 1, PASMC and PAAF were collected from the lungs of healthy donors and analyzed for transcriptomics and proteomics; in Figure 1A, it can be taken as if both cells from IPAH patients were also analyzed, but this is not reflected in the results. In Figure1D, immunostaining of normal lungs confirms the localization of PASMC and PAAF markers found by transcriptomics. The authors describe a strong, but not perfect, correlation between the transcriptomics and proteomics data from Figure S1, but the gene names of each cellular marker they found should also be listed. In addition, the authors have observed the expression of markers characteristic of PASMC and PAAF in pulmonary vessels of healthy subjects by IH, but is there any novelty in these markers? Furthermore, are the expression sites of these markers altered in IPAH patients?

In the revised manuscript we have adjusted the schematic to reflect the fact that only donor cells are compared in Figure 1. We additionally provide a correlation of cell type markers between proteomic and transcriptomic data sets for those molecules that are detected in both datasets (Figure S1B).

We provide clarification on the novelty aspect in the result section: “Some of the molecules were previously associated with predominant SMC, such as RGS5 and CSPR1 (Crnkovic et al., 2022; Snider et al., 2008), or adventitial fibroblast, such as SCARA5, CFD and MGST1 (Crnkovic et al., 2022; Sikkema et al., 2023) expression”. Except for RGS5, expression and localization of other markers in IPAH was previously unknown.

The conservation of expression sites for reported markers was validated in IPAH in the revised manuscript (Figure 2D), with IGFBP5 showing dual localization in both cell types. Moreover, results in Figure 1D, 1E and 2D support the validity of omics findings and preservation of key markers during passaging.

(2) In Figure 2, the authors compare PASMC and PAAF derived from IPAH patients and donors. The results show that transcriptomics and proteomics changes are clearly differentiated by cell type and not by pathological state. In the pathological state, transcriptional changes are more pronounced. The GO analysis of the factors that showed significant changes in each cell type is shown in Figure 2E, but the differences between the GO analysis of the transcriptomics and proteomics results are not clearly shown. The reviewer believes that the advantages of a combined analysis of both should be indicated. Also, in Figure 2G, the GAG content in PA appears to be elevated in only 3 cases, while the other 5 cases appear to be at the same level as the donor; is there a characteristic change in these 3 cases? Figure 2I shows that the phenotype of PAAF changes with cell passages. Since this phenomenon would be interesting and useful to the reader, additional discussion regarding the mechanism would be desired.

We have integrated both data sets in order to achieve stronger and meaningful analysis due to weaker and uncomplete correlation between transcriptomic and protein dataset as indicated in the results section: “Comparative analysis of transcriptomic and proteomic data sets revealed a strong, but not complete level of linear correlation between the gene and protein expression profiles (Figure S1B, C). We therefore decided to use an integrative dataset and analyzed all significantly enriched genes and proteins (-log10(P)>1.3) between both cell types to achieve stronger and more robust analysis”. In general, proteomic profile showed fewer significant differences and extent of change was lesser compared with transcriptomics, likely due to technical limitations of the method and sensitivity, testified by the complete lack of top transcriptomic molecules (RGS5, ADH1C, IGFBP5, CFD, SCARA5) in the protein dataset.

To strengthen the findings of increased GAG in IPAH pulmonary arteries, we have performed compartment-specific, quantitative image analysis of Alcian blue staining on additional donor and patient samples (n=10 for each condition). The new analysis totaling around 40 PA confirmed significantly increased deposition of GAG in IPAH pulmonary arteries.

We have addressed the issue of phenotypic change with prolonged cell culture in the revised manuscript by systematically comparing enrichment for biological processes between fresh (Crnkovic et al., 2022: GSE210248), very early (this study: GSE255669) and later passage cells (Chelladurai et al., 2022: GSE144932; Gorr et al., 2020: GSE144274). We observed cell type differences in the rate of change of phenotypic features, with PAAF showing faster shift early on during culturing that could for some of the features be due to isolation from immunomodulatory environment or presence of hydrocortisone supplement in the PAAF cell media. These points have been described in the revised results section and mentioned in the discussion.

(3) The authors claim that one feature of this paper is the use of "very early passage (p1)" of pulmonary artery smooth muscle cells (PASMC). Since there are other existing (previouly reported) data that are publicly available, such as RNA-seq data using cells with 2-4 cell passages, it may be possible to show that fewer passages are better in primary culture by comparing the data presented in this paper.

Following reviewers’ comments, we have performed systematic comparison (Crnkovic et al., 2022: GSE210248), very early (this study: GSE255669) and later passage cells (Chelladurai et al., 2022: GSE144932; Gorr et al., 2020: GSE144274). in the revised manuscript in order to comprehensively address the issue and define changes occurring as a result of prolonged in vitro conditions (Figure 3H). The results showed that the expression profile of early passage cells retains some of the key phenotypic features displayed by cells in their native environment, with PASMC displaying a more gradual loss of phenotypic characteristics compared to PAAF. Interestingly, PAAF displayed a striking inverse enrichment for inflammatory/NF-kB signaling between fresh and cultured PAAF, which could potentially be caused by the hydrocortisone supplement in the PAAF cell media or due to the isolation from its highly immunomodulatory enviroment. These points have been described in the revised results section and mentioned in the discussion.

(4) The authors describe a study characterized by decreased expression of "cytoskeletal contractile elements" in pulmonary artery smooth muscle cells (PASMC) derived from patients with IPAH. What are the implications of this result, and does it arise from the use of smooth muscle in patients resistant to pulmonary artery smooth muscle dilating agents? A discussion on this issue needs to be made in a way that is easy for the reader to understand.

The reviewer raises an interesting point regarding the loss the contractile markers and response to vasodilating therapy. We would speculate that isolated decrease in contractile machinery, without concomitant change in ECM and other PASMC features, would dampen both the contraction and relaxation properties of the single PASMC, affecting not only its response to dilating agents, but also to vasoconstrictors. Clinical consequences and responsiveness to dilating agents are more difficult to predict, since the vasoactive response would additionally depend on mechanical properties of the pulmonary artery defined by cellular and ECM composition. Nevertheless, we believe that decreased expression of contractile machinery reflects an intrinsic, “programmed” response of SMC to remodeling, rather than vasodilator therapy-induced selection pressure, since similar phenotypic change is observed in SMC from systemic circulation and in various animal models without exposure to PAH medication. These considerations have been included in the revised discussion section.

(5) There are a lot of secreted proteins that increase or decrease in Figure 6G, but there is scant reason to focus on PTX3 and HGF among them. The authors need to elaborate on the above issue.

We regret the lack of clarity and provide improved explanation of the ligand selection strategy in the revised manuscript. In order to prioritize the potential hits, we first used hierarchical clustering to group co-regulated ligands into smaller number of groups. We then prioritized for the ligands that lacked or had limited information with respect to IPAH. Based on these results, we analyzed the effect of three additional ligands on PASMC cell state marker expression (Figure S8). This additional data supported the initial focus on PTX3 and HGF.

Minor comments:(1) Regarding the number of specimens used in the Result, it would be more helpful to the reader if the number of samples were also mentioned in the text.

We have included the number of used samples in manuscript text.

(2) There is no explanation of what R2Y represents in Figure 2B. This reviewer is not able to understand the statistical analysis of Figure 2H. The detailed results should be explained.

We apologize for the oversight in labeling of Figure 2B and modify the figure legend: “Orthogonal projection to latent structures-discriminant analysis (OPLS-DA) T score plots separating predictive variability (x-axis), attributed to biological grouping, and non-predictive variability (technical/inter-individual, y-axis). Monofactorial OPLS-DA model for separation according to cell type or disease. C Bifactorial OPLS-DA model considering cell type and disease simultaneously. Ellipse depicting the 95% confidence region, Q2 denoting model’s predictive power (significance: Q2>50%) and R2Y representing proportion of variance in the response variable explained by the model (higher values indicating better fit)”.

We also modified figure legend wording for the analysis in Figure 2H (new Figure 3E) to clarify the independent factors whose interaction was investigated using 3-way ANOVA: “Interaction effects of stimulation, cell type, and disease state on cellular proliferation were analyzed by 3-way ANOVA. Significant interaction effects are indicated as follows: * for stimulation × cell type interactions and # for cell type × disease state interactions (both *, # p<0.05)”.

(3) In Figure 3, the authors examined whether there were molecular abnormalities common to IPAH-PASMC and IPAH-PAAF and found that the number of commonly regulated genes and proteins was limited to 47. Further analysis of these regulators by STRING analysis revealed that factors related to the regulation of apoptosis are commonly altered in both cells. On the other hand, the authors focused on mitochondria, as SOD2 is downregulated, and found an increase in ROS production specific to PASMC, indicating that mitochondrial dysfunction is common to PASMC and PAAF in IPAH, but downstream phenomena are different between cell types. Factors associated with apoptosis regulation have been found to be both upward and downward regulated, but the actual occurrence of apoptosis in both cell types has not been addressed.

We have performed TUNEL staining on FFPE lung tissue from donors and IPAH patients that revealed apoptosis as a rare event in both conditions in PASMC and PAAF. Therefore, no meaningful quantification could be conducted. An example of pulmonary artery where rare positive signal in either PAAF or PASMC could be found is provided in Figure 4H.

Unfortunately, association of a particular gene with a pathway is by default arbitrary and potentially ambiguous. In particular, factors identified as associated in apoptosis are also involved in regulation of inflammatory signaling (BIRC3, DDIT3) and amino acid metabolism (SHMT1). Nevertheless, mitochondria represent a crucial cellular hub for apoptosis regulation and, as shown in the current study, display significant functional alterations in IPAH in both cell types, aligning with reduced mitochondrial superoxide dismutase (SOD2) expression.

(4) The meaning of the gray circle in Figure 3C should be clarified. Similarly, the meaning of the color in Fig. 3D should be clearly explained. In Figure 3E-G, each cell is significantly different from 18-61 cells, and the number of each cell and the reason should be described.

We regret the confusion and provide better explanation of the figure legend: “gray nodes representing their putative upstream regulators”, “with color coding reflecting the IPAH dependent regulation”. In the revised Figure panels 4E-G (old 3E-G) we provide the exact number of cells measured in each condition. Although we tried to have comparable cell confluency at the time of measurement, different proliferation rates between cells from different cell type and condition led to different number of measured cells per donor/patient.

(5) In Figure 4, the authors focus on factors that vary in different directions between cells, revealing fingerprints of molecular changes that differ between cell types, particularly IPAH-PASMC, which acquires a synthetic phenotype with enhanced regulation of chemotaxis elements, whereas IPAH-PAAF, a fast cycling cell characteristics. Next, focusing on the ECM components that were specifically altered in IPAH-PASMC, Nichenet analysis in Figure 5 suggested that ligands from PAAF may act on PASMC, and the authors focused on integrin signaling to examine ECM contact and changes in cell function. The results indicate that adhesion to laminin is poor in PASMC. Although no difference was observed between donor and IPAH PASMCs, a discussion of the reasons for this would be desired and helpful to the readers.

Both donor and IPAH PASMCs respond similarly to laminin. However, our key finding is the downregulation of laminin in IPAH PAAF, which likely leads to a skewed laminin-to-collagen ratio and altered ECM composition in remodeled arteries. This shift in the ECM class results in altered PASMC behavior, affecting both donor and IPAH cells similarly. In the revised manuscript, we demonstrate that PASMC largely retain the expression pattern of integrin subunits that serve as high-affinity collagen and laminin receptors, with higher levels compared to PAAF (Figure 6F, G). Furthermore, we speculate that the distinct cellular phenotypic responses to collagen versus laminin coatings may arise from different downstream signaling pathways activated by the various integrin subunits (Nguyen et al., 2000). These considerations have been included in the revised discussion: “The comparable responses of donor and IPAH PASMC likely result from their shared integrin receptor expression profiles. Meanwhile, ECM class switching engages different high-affinity integrin receptors, which activate alternative downstream signaling pathways (Nguyen et al., 2000) and lead to differential responses to collagen and laminin matrices. We thus propose a model in which laminins and collagens act as PAAF-secreted ligands, regulating PASMC behavior through their ECM-sensing integrin receptors.”

(6) Since Figure 3B and Figure 4A seem to show the same results, why not combine them into one?

Indeed, these figure panels show the same results, but the focus of the investigations in each Figure is different. We therefore opted to keep the panels separate for better clarity and logical link to other panels in the same figure

(7) In Figure 6, the interaction analysis of scRNAseq data with respect to signaling between PASMC and PAAF was performed using Nichenet and CellChat, showing that signaling from PAAF to PASMC is biased toward secreted ligands and that a functionally relevant set of soluble ligands is impaired in the IPAH state. From there, they proceeded with co-culture experiments and showed that co-culture healthy PASMC with PAAF of IPAH patients abolished PASMC markers in the healthy state. Furthermore, the authors attempted to identify ligands that induce functional changes in PASMCs produced from IPAH PAAFs and found that HGF is a factor that downregulates the expression of contractile markers in PASMCs. Further insights may be gained by co-culturing IPAH-derived cells in co-culture experiments. Also, no beneficial effect of pentraxin3 was found in Figure 6H. The authors should examine the effect of pentraxin3 on PASMC cells derived from IPAH patients, rather than healthy donors.

We tested the influence of IPAH-PASMC on donor-PAAF and found no effect on the expression of the selected markers. We thank the reviewer for the suggestion to conduct the experiments on IPAH-PASMC. The new data show that both PTX3 and HGF have a significant effect, but differential effect on IPAH-PASMC as compared to donors-PASMC. Whereas PTX lacks effect on donor PASMC, it leads to downregulation of some of the contractile markers in IPAH PASMC, while HGF upregulates VCAN synthetic marker in IPAH PASMC. These results are now included in Figure 7H.

**Reviewer #2 (Recommendations For The Authors):**
The authors should double-check for grammar and typos in the manuscript. I caught a few such as "therefor" and others, but there could be more.

We thank the reviewer for the effort and time in reading and evaluating the manuscript. To the best of our knowledge, we have corrected the grammatical errors in the revised manuscript.